# Differentiable Registration of Images and LiDAR Point Clouds with VoxelPoint-to-Pixel Matching

**Junsheng Zhou**[1*]    **Baorui Ma**[1,2*]    **Wenyuan Zhang**[1]    **Yi Fang**[3]
**Yu-Shen Liu**[1†]    **Zhizhong Han**[4]

School of Software, Tsinghua University, Beijing, China[1]
Beijing Academy of Artificial Intelligence[2]   New York University Abu Dhabi, Abu Dhabi, UAE[3]
Department of Computer Science, Wayne State University, Detroit, USA[4]
`zhoujs21@mails.tsinghua.edu.cn`  `brma@baai.ac.cn`  `zhangwen21@mails.tsinghua.edu.cn`
`3mmyfang@nyu.edu`  `liuyushen@tsinghua.edu.cn`  `h312h@wayne.edu`

## Abstract

Cross-modality registration between 2D images from cameras and 3D point clouds from LiDARs is a crucial task in computer vision and robotic. Previous methods estimate 2D-3D correspondences by matching point and pixel patterns learned by neural networks, and use Perspective-n-Points (PnP) to estimate rigid transformation during post-processing. However, these methods struggle to map points and pixels to a shared latent space robustly since points and pixels have very different characteristics with patterns learned in different manners (MLP and CNN), and they also fail to construct supervision directly on the transformation since the PnP is non-differentiable, which leads to unstable registration results. To address these problems, we propose to learn a structured cross-modality latent space to represent pixel features and 3D features via a differentiable probabilistic PnP solver. Specifically, we design a triplet network to learn VoxelPoint-to-Pixel matching, where we represent 3D elements using both voxels and points to learn the cross-modality latent space with pixels. We design both the voxel and pixel branch based on CNNs to operate convolutions on voxels/pixels represented in grids, and integrate an additional point branch to regain the information lost during voxelization. We train our framework end-to-end by imposing supervisions directly on the predicted pose distribution with a probabilistic PnP solver. To explore distinctive patterns of cross-modality features, we design a novel loss with adaptive-weighted optimization for cross-modality feature description. The experimental results on KITTI and nuScenes datasets show significant improvements over the state-of-the-art methods. The code and models are available at https://github.com/junshengzhou/VP2P-Match.

## 1   Introduction

Image-to-Point Cloud registration finds the rigid transformation (e.g. translation and rotation) that aligns the projection of a LiDAR frame represented as a point cloud to the reference image captured by cameras. The key here is to determine the extrinsic parameters of the camera with respect to the reference frame of LiDAR. It plays an important role in autonomous driving, 3D computer vision, augmented/virtual reality, etc. Despite a plethora of approaches have explored the same-modality registration tasks (e.g. Image-to-Image [50, 13] and Point Cloud-to-Point Cloud registration [25, 1]) and achieve promising results, few researches have shown convincing performances on cross-modality registration between images and point clouds due to its inherent difficulty.

---

[*]Equal contribution.[†]The corresponding author is Yu-Shen Liu.

37th Conference on Neural Information Processing Systems (NeurIPS 2023).

Previous methods use MLPs [34, 35] on point clouds and CNNs [19, 23] on images separately with contrastive losses to learn distinctive features, and try to establish 2D-3D correspondences by matching the learned features. However, they fail to learn a structured latent space shared by 2D and 3D data due the calculation differences in MLPs or CNNs, which leads to different feature domains.

To solve these issues, we propose a novel framework to learn a structured cross-modality latent space for robust 2D-3D feature matching, where 2D elements are represented as pixels and the 3D elements are represented as the combination of voxels and points. Specifically, we design a triplet network to learn VoxelPoint-to-Pixel matching for cross-modality registration, where both the voxel and pixel branch are designed based on CNNs to operate spatially-local convolutions on voxels/pixels represented in grids, and the point branch is integrated to regain the information lost during voxelization.

To learn a distinctive latent space where the correct 2D-3D correspondences can be guaranteed with cross-modality feature matching, we propose a novel loss with adaptive-weighted optimization which allows to learn distinctive 2D-3D correspondences by optimizing positive and negative matches in a robust self-paced manner. Since the LiDAR point cloud is captured around the car with a $360^o$ perception, and the image is captured eyes forward, there is a large range of outliers on both modalities (e.g. points/voxels behind the car and sky pixels), between which there is no explicit correspondences. To handle the outliers, we design a detection strategy to predict the probability of lying in the intersection region for each 2D/3D elements, and remove the outlier regions on both modalities before inferring 2D-3D correspondences.

Furthermore, previous methods [14, 36, 26] adopt Perspective-n-Point (PnP) [24, 16] solver as a post-processing to estimate poses from matching results. They merely use the pseudo supervision conducted from 2D-3D correspondences as the optimizing target during training. The insufficient supervision leads to large errors since the network has no ability to handle incorrect matching pairs which have a highly negative effect on the results. Inspired by EPro-PnP [8], we propose to train our framework in an end-to-end manner. With a differentiable probabilistic PnP solver, we can impose supervision directly on the predicted pose by minimizing the Kullback-Leibler (KL) divergence between the predicted and target pose distribution. Our main contributions can be summarized as:

- We propose a novel framework to learn Image-to-Point Cloud registration by learning a structured cross-modality latent space with adaptive-weighted optimization, through an end-to-end training schema with a differentiable PnP solver.

- We propose to represent the 3D elements as the combination of voxels and points to overcome the pattern gap between points and pixels, where a triplet network is designed to learn VoxelPoint-to-Pixel matching.

- We demonstrate our superior performance over the state-of-the-arts by conducting extensive experiments on KITTI and nuScenes datasets.

## 2 Related Work

### 2.1 Same-Modality Registration

**Image Registration.** Image-to-Image Registration is the key of structure-from-motion (SfM) [41] and SLAM [32]. Classic methods [39, 40] extract feature descriptors from image pairs with SIFT [28] or ORB [38], and establish correspondences based on the descriptor matching, where the Perspective-n-Point (PnP) [24] or Bundle Adjustment algorithm [45] can be further applied to estimate rotations and translations from the correspondences. Recently, learning-based approaches [50, 33, 13] have shown promising results on image registration. LIFT [50] replaces the key steps of SIFT to neural network layers.

**Point Cloud registration.** Traditional methods [2, 9] on point cloud registration requires a proper initialization to estimate the rigid transformation. With the rapid development of deep learning, the neural networks have shown great potential in 3D applications [34, 55, 54, 56, 48, 27, 31, 22, 53]. Recently, learning-based approaches like 3DMatch [51], PerfectMatch [18], PPFNet [12], USIP [25] have advanced a lot on point cloud registration. 3DFeat-Net [49] attempts to joint description and detection on 3D keypoints. However, both the image and point cloud registration methods depend

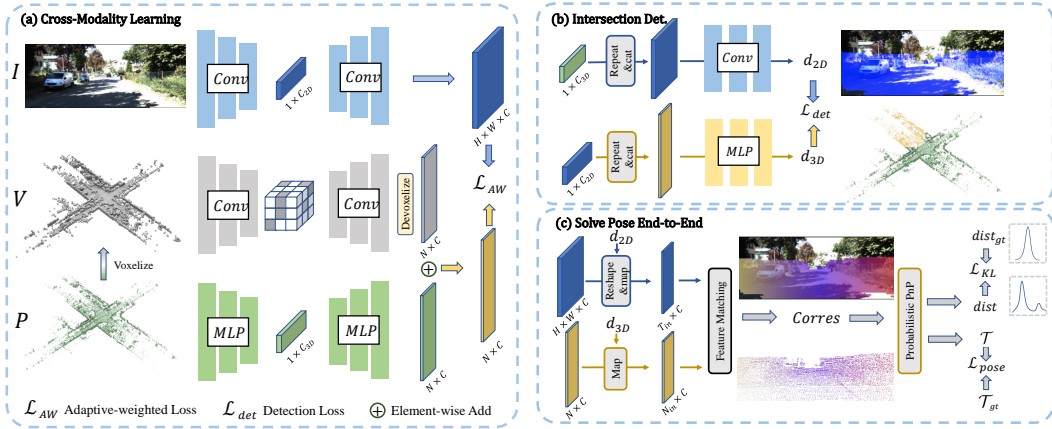

Figure 1: Overview of our method. Given a pair of mis-registered image $I$ and point cloud $P$ as input, **(a)** we first operate sparse voxelization to generate the sparse voxel $V$, and the triplet network is then applied to extract patterns from the three modalities. We represent the 2D patterns as the pixel features and the 3D patterns as the combination of voxel and point features, respectively. The adaptive-weighted loss is then used to learn distinctive 2D-3D cross-modality patterns. **(b)** We detect the intersection regions on 2D/3D space with cross-modality feature fusion. **(c)** We remove the outlier regions based on the results of intersection detection and build 2D-3D correspondences with 2D-3D feature matching, where the probabilistic PnP is then applied to predict the distribution of the extra poses to impose end-to-end supervisions with ground truth poses.

on the matching of feature descriptors designed for a specific modal, which fail dramatically on cross-modality registration.

## 2.2 Cross-Modality Registration

**Visual Localization.** A well-searched task on cross-modality registration is visual localization, which aims to estimate the 6DOF camera pose of a query image in a 3D scene model presented in point clouds. Traditional methods [44, 47] build 3D maps using SfM [41] to store 3D visual descriptors. To determine the pose of a query image, those methods match visual descriptors obtained from the query image with the ones stored in the point cloud, and apply PnP solver [24] to estimate the camera pose from 2D-3D matches. Recently, Go-Match [57] proposes a learning-based schema to solve localization with only geometry information. Some other methods [3, 7, 5] further explore the differentiability of PnP solver. But those visual localization methods only focus on locating query images on a pre-built environment, and fail to generalize to dynamic scenes captured in real-time.

Another series of methods [6, 20, 30] explore the task of LiDAR and camera self-calibration. LccNet [29] projects mis-calibrated LiDAR point clouds onto depth images, together with the images captured by cameras to regress the relative rigid transformation. However, those methods can only handle a very small mis-calibration range since a relatively large range will lead to failures on depth projection. Moreover, the self-calibration task assumes that the relative transformation between camera and LiDAR is constant and is only predicted once for each sequence, while our method focuses on the registration for each frame of the sequence which is much more difficult.

**Image-to-Point Cloud Registration.** 2D3D-MatchNet [14] detects keypoints from images and point clouds using SIFT and ISS, and feed the local patches around those 2D/3D keypoints to CNNs and PointNet. The networks are trained with triplet loss to learn cross-modality correspondences. However, the hand-crafted keypoint detectors for different modalities don't guarantee the correct matching of keypoints, which leads to a poor registration accuracy. DeepI2P [26] converts the registration problem into a classification and inverse camera projection optimization problem, but the performance is limited since the frustum classification only indicates coarse 2D-3D correspondence. Recently, CorrI2P [36] proposes to learn dense 2D-3D correspondences by matching point-wise and pixel-wise features for fine registration. However, it fails to learn a distinctive latent space for 2D and 3D data, which leads to a large ratio of incorrect correspondences. Furthermore, all the above methods can not predict the rigid transformation directly, which require a post processing (e.g. Perspective-n-Point) to estimate rigid transformation from the 2D-3D correspondences.

# 3 Method

**Problem statement.** Given a pair of 2D image $I \in \mathbb{R}^{H \times W \times 3}$ and 3D point cloud $P \in \mathbb{R}^{N \times 3}$, the cross-modality Image-to-Point Cloud registration is to predict the rigid transformation $\mathcal{T} = [R|t]$ in 3D space that can be used to align the projection of point cloud $P$ to image $I$, where $t \in \mathbb{R}^3$ is the translation vector and $R \in SO(3)$ is the rotation matrix.

In this section, we first introduce the detailed framework of our proposed VoxelPoint-to-Pixel Matching for learning a structured cross-modality latent space. Next, a novel loss with adaptive-weighted optimization is proposed for learning distinctive cross-modality patterns. Finally, we present the differentiable probabilistic PnP solver which drives our end-to-end learning schema. The overview of our framework is shown in Figure 1.

## 3.1 VoxelPoint-to-Pixel Matching

The first step of our framework is to obtain element-wise 2D and 3D features. We represent 2D elements as pixels and 3D elements as the combination of voxels and points. To achieve this, we design a triplet network consisting of Voxel/Point/Pixel branches as shown in Figure 1.(a).

**Triplet Network.** To design a *voxel branch* to obtain voxel-wise features $f_{voxel}$, a naive implementation is to generate dense voxels from $P$ and operate volumetric convolutions [58]. However, suffering from the cubic complexity of voxels, the resolution for voxel data is usually limited (e.g. $128^3$), which will lead to a large information loss, especially for large scale LiDAR data. To address this issue, we leverage sparse convolution [10, 43] on high-resolution sparse voxels where the empty voxels are skipped.

Although the *voxel branch* can represent powerful 3D spatial patterns with convolutions, it still suffers from the information loss during voxelization. Therefore, we integrate the *point branch* to regain the lost 3D detailed patterns by learning point features. The *point branch* is designed based on PointNet++ [35]. It operates hierarchical set abstraction with self-attention modules [52] to get global feature $g_{3D} \in \mathbb{R}^{C_{3D} \times 1}$, and the feature propagation module [35] is further applied to learn point-wise feature $f_{point} \in \mathbb{R}^{N \times C}$. The *pixel branch* is a convolutional U-Net [37], where the ResNet [19] is used as the basic layers to get global 2D image feature $g_{2D} \in \mathbb{R}^{C_{2D} \times 1}$, and the upsampling module with skip-connection is further applied to get pixel-wise feature $f_{pixel} \in \mathbb{R}^{H \times W \times C}$.

**2D-3D Feature Matching**. We represent 3D elements as the combination of voxels and points. To achieve this, we first transform the voxel feature to the point-wise voxel representation $f'_{voxel} \in \mathbb{R}^{N \times C}$ by interpolating each point-location with its 8 neighbor voxel grids using trilinear interpolation. Thus, the 3D features $f_{3D}$ can be donated as:

$$f_{3D} = f_{point} + f'_{voxel}, \in \mathbb{R}^{N \times C}, \tag{1}$$

and the 2D features $f_{2D} \in T \times C$ can be achieved by reshaping pixel-wise features $f_{pixel}$ where $T = H \times W$. After being processed with the embedding heads, the 2D and 3D features can be mapped into the same latent space, where we establish correspondences according to cosine similarities between $f_{2D}$ and $f_{3D}$.

**Comparison to Point-to-Pixel Matching.** Due to the different characteristics between 3D point cloud and 2D images, it is extremely difficult to operate the same feature extractors on both modals. Existing approaches [14, 36] learn point-to-pixel matching with MLP-based point network and CNN-based image network, which leads to different feature domains.

We find that although the huge domain gap between points and pixels, while the voxels and pixels share great similarities. Therefore, we introduce the voxel branch with spare convolution to capture spatially-local patterns as 2D convolution operated on pixel branch. Since both the 2D and 3D features are processed by convolutions,

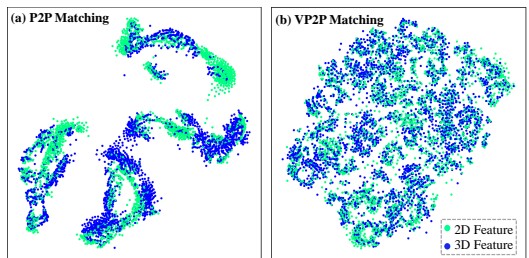

Figure 2: The t-SNE visualization of learned latent space with Point-to-Pixel (P2P) and VoxelPoint-to-Pixel (VP2P) Matching.

they share similar characteristics in feature space, leading to a structured shared latent space for both 2D and 3D features.

We visualize the learned latent space of Point-to-Pixel Matching and our proposed VoxelPoint-to-Pixel Matching with t-SNE [46] to convert features to 2D space through cosine similarities in Figure 2. The latent space learned by P2P Matching is extremely irregular with large gaps between several clusters and some space only contains the features of a single modality. In contrast, VP2P Matching leads to a structured cross-modality latent space where the features are evenly distributed throughout the space. See appendix for more analysis.

**Intersection Detection**. Since the images and LiDAR point clouds are captured in quite different ways, there is a large range of outliers on both modalities where no correspondences can be founded. We define the intersection region as the overlap between the 2D projection of a LiDAR point cloud using ground truth camera parameters and the reference image. To handle the outlier regions, we design a detection strategy as shown in Figure 1(b) to predict the probability of lying in the intersection region for each 2D/3D elements, and remove the outlier regions on both modalities before inferring 2D-3D correspondences.

For detecting the outlier region in 3D space, we first repeat the 2D global feature $g_{2D}$ and concatenate it with the 3D element-wise feature $f_{3D}$ as $[g_{2D} : f_{3D}]$, where ":" donates the feature concatenation. Then several MLPs $\phi$ are used to learn the probability $d_{3D}^i$ of a point $p^i$ to lie in the 2D-3D intersection region as: $d_{3D}^i = \phi(f_{3D} : g_{2D}), i \in [1, N]$ , vice versa for detecting 2D space outliers. We define a threshold $\sigma$ where 2D/3D elements with probabilities smaller than $\sigma$ will be considered as outliers.

To learn the detection, we sample $Z_{in}$ pairs of pixels and points with probability $d$ from the intersection region and $Z_{out}$ pairs of pixels and points with probability $\hat{d}$ from the outlier regions. The detection loss is then formulated as:

$$\mathcal{L}_{det} = \frac{1}{Z_{out}} \sum_{i=0}^{Z_{out}} (\hat{d}_{2D}^i + \hat{d}_{3D}^i) - \frac{1}{Z_{in}} \sum_{j=0}^{Z_{in}} (d_{2D}^j + d_{3D}^j) \tag{2}$$

## 3.2 Adaptive-Weighted Optimization

To explore distinctive patterns, various optimization strategies like contrastive loss and triplet loss are widely used in 2D or 3D tasks. However, these formulations treat each pair of samples equally, which leads to ambiguous convergence, especially in the difficult situation of 2D-3D feature matching. Inspired by the Circle Loss [42], we design a flexible optimization strategy with adaptive weighting for learning a distinctive cross-modality latent space where we can establish 2D-3D correspondences more accurately.

Given a set of 2D-3D pairs $K = \{\alpha^i, \beta^i\}, i \in [1, v]$, which are sampled from the intersection region. We separate them into positive pairs $k_p$ and negative pairs $k_n$ determined by whether the distance between the location of $\alpha^i$ and the projection of $\beta^i$ on the image is larger than a radius $r$ or not. We define the positive similarity for a positive pair $k_p^i$ as:

$$s_p^i = f_{2D}^i \cdot f_{3D}^i = \sum_c f_{2D}^{ic} f_{3D}^{ic}, \tag{3}$$

where $c$ is the channel number of features. And vice versa for negative pairs $s_n$.

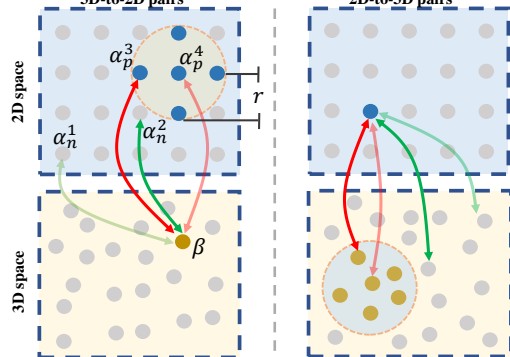

Figure 3: The illustration of Adaptive-weighted optimization.

To avoid ambiguous convergence and allow flexible optimization, we design adaptive weighting factors for positive and negative pairs as $\rho_p = \Psi(\gamma(1 - s_p^j + m))$ and $\rho_n = \Psi(\gamma(s_n^k - m))$, in which $\gamma$ is a scale factor, $m$ is a margin for better similarity separation and $\Psi$ is the function to detach $\rho_p$ and $\rho_n$ from gradients to serve as weights for optimization. The adaptive-weighted loss is then derived as:

$$\mathcal{L}_{AW} = \log[1 + \sum_j \exp^{\rho_p(1 - s_p^j + m)} \sum_k \exp^{\rho_n(s_n^k - m)}]. \tag{4}$$

We illustrate the adaptive-weighted optimization in Figure 3. The 3D space is achieved by projecting points into image space. Given a 3D point $\beta$, we define two negative pairs $\{\alpha_n^1, \beta\}$ and $\{\alpha_n^2, \beta\}$ where $\alpha_n^1$ and $\alpha_n^2$ lies outside the safe radius $r$. It is obviously that the negative pair $\{\alpha_n^2, \beta\}$ is much harder to learn since $\alpha_n^2$ lies near the positive region and some samples around $\alpha_n^2$ is from positive pairs, while $\{\alpha_n^1, \beta\}$ is easier since all the samples around $\alpha_n^1$ is also negative. And vice versa for the positive pairs. Optimizing all the samples with the same weight will lead to ambiguous convergence at the regions containing both negative and positive samples, i.e., it will be difficult to distinct the negative sample $\alpha_n^2$ with positive samples. Driven by this analysis, we optimize the patterns with adaptive weighting to force the network focus more on the harder samples (indicated by the color shading above).

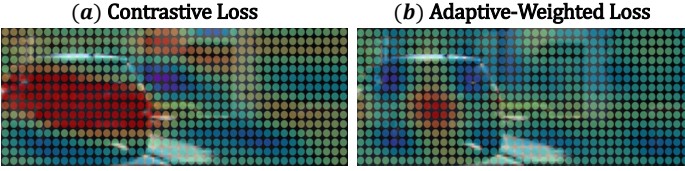

(*a*) Contrastive Loss      (*b*) Adaptive-Weighted Loss

Figure 4: The cosine similarity visualization with different losses. We select one point $p$ at the rear of the car from the point cloud and compute the cosine similarity between $p$ and each pixel in the reference image. The colors from blue to red indicate the increase of similarities.

We visually compare the feature matching results of our proposed adaptive-weighted optimization with the contrastive loss as used in CorrI2P [36] in Figure 4, where our proposed loss shows significantly superior performance in learning a distinctive cross-modality latent space.

### 3.3 Differentiable PnP Solver

**Correspondences Establishing.** To establish correspondences, we first remove the outlier regions on both modalities with intersection detection and leverage the nearest neighbor principle on the cross-modality latent space for 2D-3D feature matching. Due to the different density of points and pixels, they do not have completely one-to-one correspondences. To avoid ambiguous matching, we establish the 2D-3D correspondence $X^i$ by searching for the point coordinate $P^z$ with the largest similarity in cross-modality latent space for each 2D pixel coordinate $I^i, i \in [1, T_{in}]$, $T_{in}$ is the number of pixels which lie in the intersection region. The operation achieved by $\arg\max$ to find $P^z$ is inherently non-differentiable, to train our framework end-to-end, we leverage *Straight-Through Gumbel Estimator* [21] to estimate the gradient for backward optimization.

**Probabilistic PnP.** Given a set of correspondences $X = \{I^i, P^i\}, i \in [1, T_{in}]$, the PnP problem [24] is to search for an optimal pose $\mathcal{T} = [R|t]$ that minimizes the reprojection error:

$$\arg\max_{\mathcal{T}} \frac{1}{2} || \underbrace{\pi(RP^i + t) - I^i}_{w_i(\mathcal{T})} ||^2, \tag{5}$$

where $\pi$ is the projection function and $w^i(\mathcal{T})$ is the reprojection error at $X^i$. Inspired by Epro-PnP [8], we solve the non-differentiable PnP problem by interpreting the output as a probabilistic distribution:

$$p(\mathcal{T}|X) = \frac{\exp - \frac{1}{2} \sum_{i=1}^{T_{in}} ||w^i(\mathcal{T})||^2}{\int \exp - \frac{1}{2} \sum_{i=1}^{T_{in}} ||w^i(\mathcal{T})||^2 \mathrm{d}\mathcal{T}}. \tag{6}$$

The KL divergence loss is then computed to minimize the distance between the predicted pose distribution and ground truth pose distribution:

$$\mathcal{L}_{KL} = \frac{1}{2} \sum_{i=1}^{T_{in}} ||w^i(\mathcal{T}_{gt})||^2 + \log \int \exp - \frac{1}{2} \sum_{i=1}^{T_{in}} ||w^i(\mathcal{T})||^2 \mathrm{d}\mathcal{T}, \tag{7}$$

where the first term is the the reprojection error at ground truth pose, and the second term is the integral of reprojection error at predicted pose distribution. In practice, we leverage Monto Carlo strategy to collect samples to approximate the integration with Adaptive Multiple Importance Sampling algorithm [11].

Except for conducting supervisions on the probabilistic distribution, we further estimate the exact pose $\mathcal{T}' = [R'|t']$ by solving Eq. (5) with an iterative PnP solver based on Gauss-Newton (GN) algorithm, and compute the pose loss as:

$$\mathcal{L}_{pose} = 2 - 2(R'^T R_{gt})^2 + ||t' - t_{gt}||_2^2, \tag{8}$$

which can also participate on the optimization since the iterative part in GN algorithm is differentiable.

Table 1: Registration accuracy on the KITTI and nuScenes datasets. Lower is better for RTE and RRE, higher is better for Acc.

| Method | KITTI | | | nuScenes | | |
|---|---|---|---|---|---|---|
| | RTE(m)↓ | RRE($^o$)↓ | Acc.↑ | RTE(m)↓ | RRE($^o$)↓ | Acc.↑ |
| Grid Cls. + PnP [26] | $3.64 \pm 3.46$ | $19.19 \pm 28.96$ | 11.22 | $3.02 \pm 2.40$ | $12.66 \pm 21.01$ | 2.45 |
| DeepI2P (3D) [26] | $4.06 \pm 3.54$ | $24.73 \pm 31.69$ | 3.77 | $2.88 \pm 2.12$ | $20.65 \pm 12.24$ | 2.26 |
| DeepI2P(2D) [26] | $3.59 \pm 3.21$ | $11.66 \pm 18.16$ | 25.95 | $2.78 \pm 1.99$ | $4.80 \pm 6.21$ | 38.10 |
| CorrI2P [36] | $3.78 \pm 65.16$ | $5.89 \pm 20.34$ | 72.42 | $3.04 \pm 60.76$ | $3.73 \pm 9.03$ | 49.00 |
| Ours | $\mathbf{0.75 \pm 1.13}$ | $\mathbf{3.29 \pm 7.99}$ | **83.04** | $\mathbf{0.89 \pm 1.44}$ | $\mathbf{2.15 \pm 7.03}$ | **88.33** |

## 3.4 Implementation Details

We set the channel dimension $C$ of 2D/3D feature to 64, and set $C_{2D}$ and $C_{3D}$ for the 2D/3D global feature both to 512. We set the margin $m$ to 0.25, the scale factor $\gamma$ to 32 and the safe radius $r$ to 1 pixel. To enhance the representation ability of the voxel branch, we keep the point transformation pipe used in SPVNAS [43]. And the probability threshold $\sigma$ in intersection detection is set to 0.95. Although the GN-based PnP solver used in Sec. 3.3 can solve exact pose while remain differentiable, its iterative solution is very time-consuming. For efficient registration, we leverage the EPnP [24] with $O(n)$ time complexity during inference since we do not focus on the differentiability at inference time. The RANSAC [15] is applied with EPnP for more robust registrations.

# 4 Experiments

## 4.1 Dataset

We evaluate our performance on Image-to-Point Cloud registration task on two wildly used benchmarks KITTI and nuScenes. On both dataset, the images and point clouds are captured simultaneously with 2D cameras and 3D LiDARs.

**KITTI Odometry [17].** We generate the image-point cloud pairs from the same data frame of 2D/3D sensors. We follow previous works [26] to utilize the 0-8 sequences for training, and 9-10 for testing. The mis-registration transformation is conducted with a 2D translation on the ground within $\pm10$ and a rotation around the up-axis with no limited range. We downsample the image resolution to $160 \times 512$ and the point cloud size to 40960 for training and testing.

**nuScenes [4].** The image-point cloud pairs are generated by official SDK where the point cloud is accumulated from the nearby frames and the image is from the current frame. We follow the official data spilt of nuScenes to utilize 850 scenes for training, and 150 scenes for testing. The mis-registration transformation is conducted in a similar way as the one in KITTI dataset. We downsample the image resolution to $160 \times 320$ and the point cloud size to 40960, respectively.

## 4.2 Baselines and Metrics

We compare our method with the state-of-the-art methods DeepI2P [26] and CorrI2P [36]:

**1) Grid Cls. + PnP.** The grid classification setting is a baseline approach proposed in DeepI2P to divide the image into $32 \times 32$ grids and learn to classify each 3D point to a unique 2D grid with a neural network. The EPnP with RANSAC is then applied to predict the rigid transformation.
**2) Frus.Cls. + Inv.Proj.** DeepI2P proposes to perform the frustum classification with the inverse camera projection to obtain final rigid transformation. We report their results with 2D and 3D inverse camera projection as DeepI2P(2D) and DeepI2P(3D).

**3) CorrI2P.** CorrI2P is the latest work on Image-to-Point Cloud registration which learns dense correspondences for image-point cloud pairs, and the EPnP with RANSAC is applied to predict the rigid transformation.

**Metrics.** We follow DeepI2P to evaluate the registration performance with average Relative Translational Error (RTE) and average Relative Rotation Error (RRE). We do not follow CorrI2P to remove the image-point cloud samples with large errors before averaging, which in our opinion, is an unsuit-

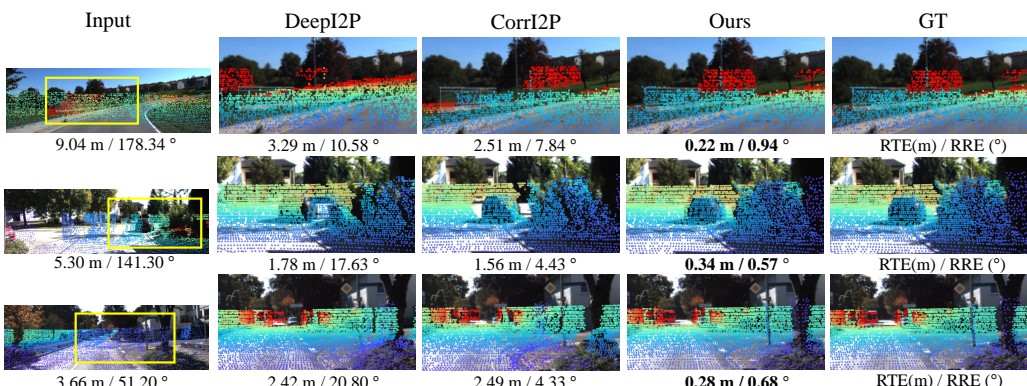

| Input | DeepI2P | CorrI2P | Ours | GT |
|---|---|---|---|---|
| 9.04 m / 178.34 ° | 3.29 m / 10.58 ° | 2.51 m / 7.84 ° | **0.22 m / 0.94 °** | RTE(m) / RRE (°) |
| 5.30 m / 141.30 ° | 1.78 m / 17.63 ° | 1.56 m / 4.43 ° | **0.34 m / 0.57 °** | RTE(m) / RRE (°) |
| 3.66 m / 51.20 ° | 2.42 m / 20.80 ° | 2.49 m / 4.33 ° | **0.28 m / 0.68 °** | RTE(m) / RRE (°) |

Figure 5: Visual comparison of Image-to-Point Cloud registration results under KITTI dataset.

able way to report real performances. We further report the registration accuracy (Acc.) which is the proportion of fine registrations with RTE < 2m and RRE < $5^o$.

## 4.3 Registration Accuracy

**Quantitative Comparison.** The results of cross-modality registration are shown in Table 1, in which our proposed method achieves the superior performance over the compared methods on both KITTI and nuScenes dataset. Especially, the latest work CorrI2P [36] adopted a similar way as ours to establish 2D-3D correspondences with feature matching, which is the most relevant method to ours. However, our method is about 4 times better than CorrI2P in terms of RTE. The main reason is that the previous methods fails to conduct end-to-end supervisions and can not learn robust 2D-3D correspondences due to the huge domain gap. In contrast, our proposed method can learn a structured cross-modality latent space for robust 2D-3D feature matching, together with an end-to-end training schema driven by a probabilistic PnP solver. As a result, our method is able to predict highly accuracy 2D-3D correspondences and achieve better performance than its counterparts.

**Visual Comparison.** The visual comparison is shown in Figure 5. For intuitive visualization, the point cloud is projected into image space with the predicted extrinsic parameters of different approaches and the known camera internal parameters. The color indicates the distance between a point and the camera. For DeepI2P, we adopt the setting with highest accuracy, i.e. DeepI2P(2D) for visual comparison. Compared with the other methods, our method achieves better registration accuracy in different road situations. For example, on the difficult tuning situation (e.g. the $1^{st}$ and $2^{nd}$ row), both DeepI2P and CorrI2P fail to solve correct registration, where the projections of trees and cars are largely dis-matched with the corresponding pixels in the image.

## 4.4 Feature Matching Accuracy

We further provide the feature matching visualization in Figure 6. We visualize the double-side error maps by computing the matching distance on both modalities. Specifically, for 2D-to-3D matching, we search for a point with the greatest similarity in cross-modality latent space for each 2D pixel in the intersection region, and compute the error by first projecting the matching point into image space and compute the Euler distance between the projected matching point and the 2D pixel.

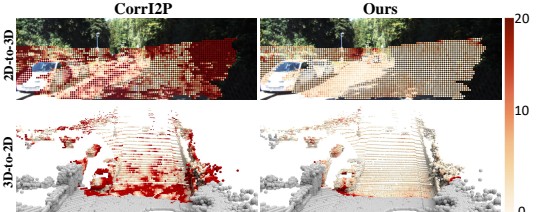

Figure 6: Feature matching error visualization.

And vice versa for 3D-to-2D matching. As shown in Figure 6, our method significantly outperforms CorrI2P [36] in both 2D-to-3D and 3D-to-2D matchings. Most of the pixels in 2D-to-3D matching or points in 3D-to-2D matching of our method can reach slight errors within 2 pixels, which demonstrates that our learned shared latent space can distinct cross-modality patterns apart and lead to an accurate feature matching for each single 2D/3D element. We also observe that some elements on the 2D/3D edges will lead to relative large errors, since it is often difficult for intersection detection to perform perfectly in the edges.

## 4.5 Efficiency Comparison

We further compare the efficiency of our method with other counterparts, where all the methods are evaluated with an NVIDIA RTX 3090 GPU and Intel(R) Xeon(R) E5-2699 CPU. The results are shown in Table 2, where our method requires much fewer parameters but gets significantly better performances. Furthermore, our method only requires $0.19s$ for network inference and pose estimation for one frame, which is about $50\times$ (or more) faster than the previous works. The reason is that DeepI2P solves the time-consuming inverse camera projection problem for pose estimation, even with a 60-fold pose initialization for avoiding crashing, and CorrI2P spends a lot time to eliminate the wrong correspondences through iteration. Therefore, all the previous works fail to be applied to real automatic driving scenarios where the low latency is required, while our method can solve poses much more efficiently. Note that our result is achieved by replacing the GN-based PnP solver with the O(n) EPnP [24] at inference time, while the time will increase to $2.38s$ with the GN-based PnP inference.

Table 2: The efficiency comparisons with other methods.

|  | DeepI2P (2D) | DeepI2P (3D) | CorrI2P | Ours |
|---|---|---|---|---|
| Model size (MB) | 100.12 | 100.12 | 141.07 | **30.73** |
| Pose Inference (s) | 23.47 | 35.61 | 8.96 | **0.19** |

## 4.6 Ablation Study

We conduct ablation studies to justify the effectiveness of each design in our method and the effect of some important parameters. We report the performance in terms of RTE/RRE/Acc. under the KITTI dataset.

**Framework designs.** We first justify the effectiveness of each design of our framework. Specifically, we develop four different variations for comparison: (1) *w/o voxel branch* is the variation removing the voxel branch from the triplet network; (2) *w/o point branch* is the variation removing the point branch from the triplet network; (3) *w/o A-W optimization* is the variation replace our adaptive-weighted optimization loss with the contrastive loss as used in CorrI2P [36]; (4) *w/o Diff. PnP* is the variation removing the end-to-end supervisions driven by the differentiable PnP solver.

Table 3: The effect of each design in our framework.

|  | RTE(m)↓ | RRE($^o$)↓ | Acc.↑ |
|---|---|---|---|
| w/o voxel branch | 1.25 ± 4.54 | 7.03 ± 10.19 | 73.53 |
| w/o point branch | 1.08 ± 3.09 | 6.91 ± 13.26 | 80.80 |
| w/o A-W optimization | 1.04 ± 2.88 | 4.81 ± 8.96 | 77.01 |
| w/o Diff. PnP | 0.83 ± 1.60 | 3.44 ± 10.02 | 82.18 |
| Full | **0.75 ± 1.13** | **3.29 ± 7.99** | **83.04** |

The results are shown in Table 3, from which we can find that our Full model achieves the best performances over all variations. Such results prove the effectiveness of each design in our framework. Moreover, by comparing *w/o voxel branch* and *w/o point branch* to the Full model, we can find that the voxel branch plays a more important role in our framework, which demonstrates that the voxel modality with similar characteristics (represented in grids) and pattern extractors (CNNs) as pixels is more suitable for learning Image-to-Point Cloud registration.

**The effect of input resolutions.** We further explore the effect of input image resolutions and point densities. Table 4 shows the performance of different settings, from which we can find that higher resolutions on both modalities lead to better results since the low resolution images will omit some visual information while the low density point clouds will lose the detailed geometries. We select the proper setting with a balance between the performance and efficiency.

Table 4: Ablations on image resolutions and point densities.

| Image | Point | RTE(m)↓ | RRE($^o$)↓ | Acc.↑ |
|---|---|---|---|---|
| 40 × 128 | 40960 | 0.92 ± 1.52 | 3.54 ± 12.08 | 82.50 |
| 80 × 256 | 40960 | 0.84 ± 1.07 | 3.65 ± 8.83 | 82.67 |
| 160 × 512 | 40960 | 0.75 ± 1.13 | 3.29 ± 7.99 | 83.04 |
| 160 × 512 | 20480 | 0.93 ± 1.33 | 3.59 ± 8.62 | 82.75 |
| 160 × 512 | 61440 | 0.72 ± 1.17 | 2.82 ± 5.37 | 83.70 |

## 5 Conclusions

In this work, we propose a novel framework to learn Image-to-Point Cloud registration with VoxelPoint-to-Pixel Matching, where we learn a structured cross-modality latent space with a novel adaptive-weighted loss. We represent the 3D elements as the combination of voxels and points to overcome the domain gap between points and pixels. Moreover, we train our framework end-to-end

by imposing supervision directly on the predicted pose distributions with a differentiable PnP solver. The extensive experiments on KITTI and nuScenes datasets demonstrate our superior performances.

# 6 Acknowledgement

This work was supported by National Key R&D Program of China (2022YFC3800600), the National Natural Science Foundation of China (62272263, 62072268), and in part by Tsinghua-Kuaishou Institute of Future Media Data.

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
