# Supplementary Material for "Differentiable Registration of Images and LiDAR Point Clouds with VoxelPoint-to-Pixel Matching"

**Junsheng Zhou**[1*] **Baorui Ma**[1,2*] **Wenyuan Zhang**[1] **Yi Fang**[3]
**Yu-Shen Liu**[1†] **Zhizhong Han**[4]
School of Software, Tsinghua University, Beijing, China[1]
Beijing Academy of Artificial Intelligence[2] New York University Abu Dhabi, Abu Dhabi, UAE[3]
Department of Computer Science, Wayne State University, Detroit, USA[4]
zhoujs21@mails.tsinghua.edu.cn brma@baai.ac.cn zhangwen21@mails.tsinghua.edu.cn
3mmyfang@nyu.edu liuyushen@tsinghua.edu.cn h312h@wayne.edu

## 1 Network Details

In this section, we introduce the detailed structures of our designed triplet network. We present the Voxel/Point/Pixel branches in Figure 1.

### 1.1 Voxel Branch

For learning spatially-local 3D features from the large-scale LiDAR point clouds efficiently, we leverage sparse convolution [7, 2] on high-resolution sparse voxels to skip the empty voxels. The network details are shown in Figure 1.(a), where each sparse convolution block consists of three sparse convolutions with residual connections. We introduce the details on data representation, (de)voxelization and feature aggregation below.

**Sparse Voxelization.** Given a point cloud $P = \{t_p^i, f_p^i\}, i \in [1, N]$ as input, where $t_p^i = \{x^i, y^i, z^i\}$ is the 3D location and $f_p^i$ is the point-wise feature. We first generate the sparse voxel representation of the point cloud $P$ as $V = \{t_v^j, f_v^j\}, j \in [1, M]$, where $t_v^j$ is the 3D coordinate of sparse voxel $v^i$ and $f_v^i$ is the voxel-wise feature of $v^j$. We then define the sparse voxelization operation to generate voxel coordinates $t_v^j$ as:

$$t_v^j = \{x_v^j, y_v^j, z_v^j\} = \{\varphi(x_p^j/l), \varphi(x_p^j)/l, \varphi(x_p^j)/l\}, \tag{1}$$

where $\varphi(\cdot)$ is the floor operation to get integer coordinates and $l$ is the voxel size which indicates the length of a voxel grid.

And the voxel feature $f_v^j$ is generated as:

$$f_v^j = \frac{1}{\hat{N}^i} \sum_{i=0}^{N} \Psi(x_v^i = x_p^i, y_v^i = y_p^i, z_v^i = z_p^i) \cdot f_p^i, \tag{2}$$

where $\Psi(\cdot)$ is the binary classifier to determine whether $t_p^i$ is inside the voxel grid $v^i$ and $\hat{N}^i$ is the number of points that located inside a non-empty voxel $v^i$.

**Sparse Convolution.** To overcome the inefficiency of volumetric convolutions, some latest works [2, 7] propose to operate sparse convolutions which skip the non-activate local regions to reduce the memory consumption. The sparse convolution is achieved by first identifying the correspondences

---

*Equal contribution.†The corresponding author is Yu-Shen Liu.

37th Conference on Neural Information Processing Systems (NeurIPS 2023).

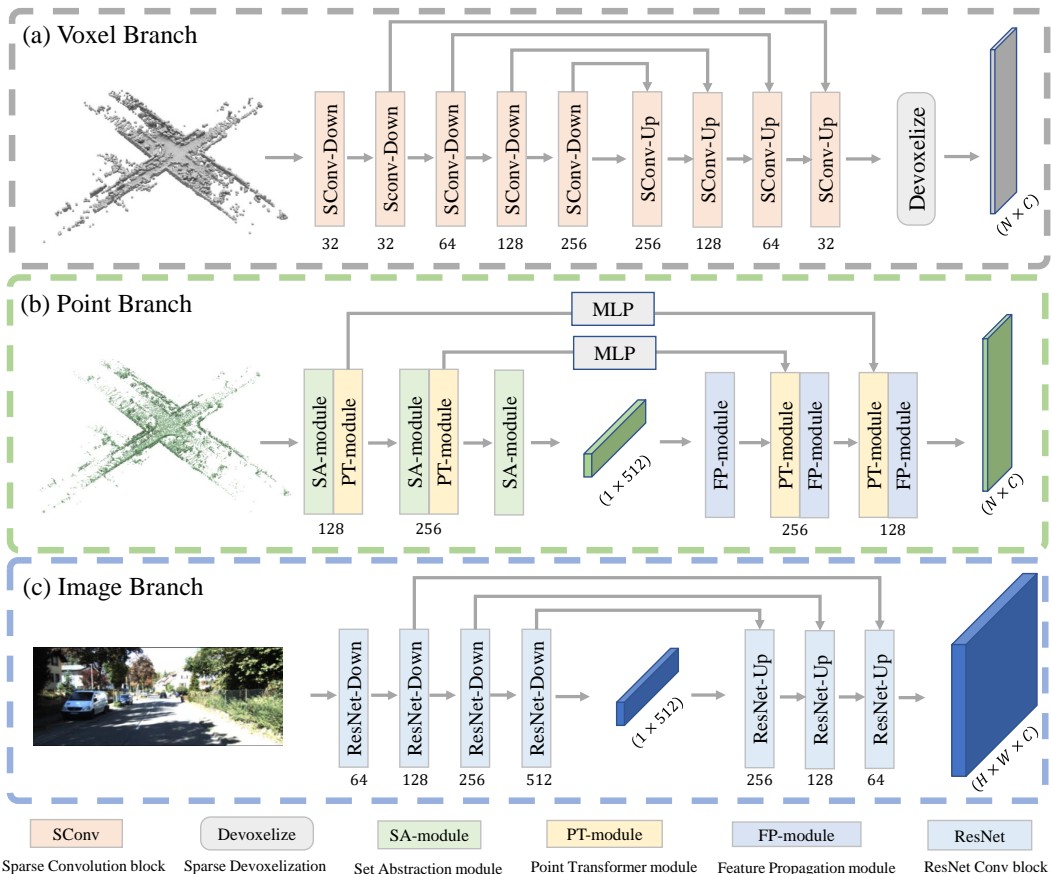

Figure 1: Detailed structures of Triplet Network for VoxelPoint-Pixel Matching.

between input and output points with a kernel map and then perform convolutions based on the map. For more details, please refer to MinkowskiNet [2].

**Sparse Devoxelization.** After aggregating sparse voxel features $f_v^j$ with a set of residual sparse convolution blocks [2], we transform the voxel feature to the point-wise voxel representation, which is fused together with point features, and serve as the final 3D features in our framework. The sparse devoxelization is achieved by interpolating each point location with its 8 neighbor voxel grids using trilinear interpolation similar to SPVNAS [7].

## 1.2   Point Branch

We provide the detailed network structure of point branch in Figure 1.(b). The point branch is designed based on PointNet++ [5] with set abstraction modules and feature propagation modules to extract patterns from point clouds, where four self-attention based PT-modules [9] are further integrated to explore more detailed patterns.

## 1.3   Pixel Branch

We provide the detailed network structure of pixel branch in Figure 1.(c). The pixel branch is a convolutional U-Net, where the ResNet is used as the basic blocks to extract image patterns. The four ResNet-Down blocks consist of [3, 4, 6, 3] residual convolutions, respectively. And each ResNet-Up block consists of a bilinear interpolation-based upsampling module and two residual convolutions.

## 2 Additional Experiments and Analysis

### 2.1 The Analysis of VP2P Matching

We provide the visualization of Point-to-Pixel (P2P) Matching and our proposed VoxelPoint-to-Pixel (VP2P) Matching with t-SNE [8] to convert features to 2D space through cosine similarities in Figure 2 of the submission. The visual comparison shows that our proposed VP2P Matching leads to a structured cross-modality latent space while P2P Matching leads to an extremely irregular latent space.

To further demonstrate the effectiveness of our proposed VP2P Matching, we provide the quantitative comparison on the latent distribution as shown in Table 1. Given a pair of image and point cloud, we first extract the 2D and 3D element-wise features with the learned P2P Matching model or VP2P Matching model, and compute the cosine similarity between the features of each pixel and each point. We search for the 3D features with greatest cosine similarity with each 2D feature in the cross-modality latent space and report the average of their similarities as "Max Similarity". We also report the standard deviation of cosine similarities as "Similarity Std."

The results in Table 1 show that VP2P Matching achieves better "Max Similarity" than P2P Matching, which demonstrate that the distribution of 2D and 3D features lies closer in the latent space learned by VP2P Matching. We also achieve a smaller "Similarity Std.", which proofs that the 2D/3D features learned by VP2P Matching have a more regular

Table 1: The comparison of VP2P and P2P matching.

|  | Max Similarity | Similarity Std. |
|---|---|---|
| P2P Matching | 0.47 | 0.27 |
| VP2P Matching | **0.54** | **0.14** |

distribution in the latent space. The conclusions are also supported by the visualization in Figure 2 of the submission, where the latent space learned by P2P Matching is extremely irregular where a large ratio of space is empty and some space only contains the features of a single modality. In contrast, VP2P Matching leads to a structured cross-modality latent space where the features are evenly distributed throughout the space.

### 2.2 KITTI vs. nuScenes

There are two main differences between the KITTI [3] and nuScenes [1] dataset. First, the point cloud of a single LiDAR frame in the KITTI dataset is dense enough for learning patterns, while that in nuScenes is relatively sparse and requires to be spliced with adjacent frames which brings noises since the adjacent LiDAR frames are captured dynamically. Second, the nuScenes dataset is a larger dataset containing about 2 times training samples than KITTI dataset. Our method shows great generality by achieving more accurate registration results on both KITTI and nuScenes datasets. However, CorrI2P shows a decline in accuracy in nuScenes, since it is unable to handle noises due to its unstable and ambiguous point-to-pixel feature matching.

### 2.3 The Effect on Safe Radius $r$

We further test the effect of safe radius $r$ which controls the range of positive samples for adaptive-weighted optimization as described in Sec.3.2 of the main paper. We report the performance in terms of RTE/RRE/Acc. under a subset of the KITTI dataset, where the sequence 0-1 is used for training and the sequence 7 is used for testing. We set the safe radius $r$ to 0.5, 1, 2 and 4 pixels, and report the performances in Table 2. We observe that a too small or too large safe radius will degenerate the performance. The reason is that a too small safe radius will lead to relative few positive pairs during training, making the network overfit to negative samples. While a too large safe radius will lead to inaccurate optimization targets since some remote samples will also be regarded as positive samples.

### 2.4 The Effect on Feature Dimension $C$

The channel dimension $C$ of cross-modality features is also a crucial factor in the network training. We report the performance in terms of RTE/RRE/Acc. under a subset of the KITTI dataset, where the sequence 0-1 is used for training and the sequence 7 is used for testing. We show the results of training our network in different number of channel dimensions $C = [16, 32, 64, 128]$ in Table 3.

| $r$ (pixel) | RTE(m)↓ | RRE($^o$)↓ | Acc.↑ | | $C$ (channel) | RTE(m)↓ | RRE($^o$)↓ | Acc.↑ |
|---|---|---|---|---|---|---|---|---|
| 0.5 | $0.88 \pm 1.68$ | $2.78 \pm 6.63$ | 89.24 | | 16 | $0.79 \pm 1.38$ | $2.64 \pm 6.35$ | 89.69 |
| 1 | $\mathbf{0.65 \pm 1.28}$ | $\mathbf{2.10 \pm 4.13}$ | **91.14** | | 32 | $0.80 \pm 1.90$ | $2.77 \pm 9.03$ | **91.87** |
| 2 | $0.69 \pm 1.15$ | $2.48 \pm 5.95$ | 90.60 | | 64 | $\mathbf{0.65 \pm 1.28}$ | $\mathbf{2.10 \pm 4.13}$ | 91.14 |
| 4 | $0.96 \pm 1.56$ | $3.16 \pm 5.88$ | 83.33 | | 128 | $1.19 \pm 2.63$ | $3.24 \pm 10.18$ | 85.65 |

Table 2: The effect of safe radius $r$.      Table 3: The effect of feature dimension $C$.

We found that a proper channel dimension around 64 will lead to the best performance. The results show that a too small channel dimension can not represent detailed 2D/3D patterns for robust feature matching, while a too large channel dimension will lead to a lot of information redundancy and make the network difficult to converge.

## 2.5 Feature Matching Accuracy

As shown in Table 4, we further provide the feature matching accuracy comparison with CorrI2P [6] which learns point-to-pixel matching for Image-to-Point Cloud registration. As described in Sec.4.4, we report the double-side error metrics by computing the matching error distances on both modalities. Specifically, for 2D-to-3D matching, we search for a 3D element (point for CorrI2P and the combination of voxel and point for our method) with the greatest similarity in cross-modality latent space for each 2D pixel in the intersection region. We then compute the error by first projecting the matching point into image space and compute the Euler distance between the projected matching point and the 2D pixel. And vice versa for 3D-to-2D matching. We report the average and standard deviation of error distances in Table 4. To further evaluate the fine matching ability of different methods, we report the matching accuracy (Acc.) which is the proportion of fine matchings with Error < 5 pixels.

The quantitative comparisons show that our method significantly outperforms CorrI2P in both 2D-to-3D and 3D-to-2D matchings in terms of error distances and accuracy. The main reason is that CorrI2P adopts the typical Point-to-Pixel matching paradigm which fails to learn robust 2D-3D correspondences due to huge domain gap between points and pixels, while we propose to learn VoxelPoint-Pixel matching for learning a structured cross-modality latent space to overcome the pattern gap and build accurate 2D-3D correspondences for cross-modality registration. The visualization comparison is shown in Figure 3.

# 3 Additional Visualizations

## 3.1 Registration Comparison on nuScenes Dataset

We provide the visualization comparisons with DeepI2P [4] and CorrI2P [6] under the nuScenes dataset in Figure 2. For DeepI2P, we adopt the setting with highest accuracy to perform the frustum classification with 2D inverse camera projection, i.e. DeepI2P (2D) for visual comparison. We also list the RTE and RRE of each method in each frame under the figures. As shown in Figure 2, our

Table 4: Feature matching accuracy on the KITTI dataset.

| Matching type | Method | Error (pixel)↓ | Acc.↑ |
|---|---|---|---|
| 2D-to-3D | CorrI2P | $5.19 \pm 4.17$ | 60.10 |
| | Ours | $\mathbf{3.43 \pm 3.42}$ | **78.85** |
| 3D-to-2D | CorrI2P | $5.28 \pm 4.30$ | 59.67 |
| | Ours | $\mathbf{3.66 \pm 3.63}$ | **76.68** |

method achieves the best registration accuracy in different road situations. Note that the cross-modality registration in nuScenes dataset is more difficult than KITTI dataset since the LiDAR point cloud in nuScenes dataset is relatively sparse and requires to be spliced with adjacent frames which brings noises.

## 3.2 Feature Matching Comparison

We further provide the feature matching visual comparison with CorrI2P in Figure 3. We visualize the error range of 0-20 pixels where the error larger than 20 pixels is set to 20 pixels. It is obvious that our method achieves better matching results in both 2D-to-3D and 3D-to-2D feature matching, where most errors of our method can be controlled within 2 pixels while CorrI2P often leads to extremely large errors.

## 3.3 Scene Fusion by Registration

We provide the scene fusion visualization results in Figure 4. The visualization is achieved by first registering each frame of LiDAR point cloud with the reference image using our method, and then fusing the registered LiDAR point clouds together with the trajectory poses provided in KITTI dataset. More specifically, assume that the camera trajectory poses is known, we use our method to register each frame of LiDAR point cloud to the reference image in this frame, and leverage the camera trajectory pose at this frame to locate the registered point cloud in the scene. After registering and locating all the LiDAR point cloud frames in the data sequence, the scene point cloud is then generated as shown in Figure 4.

Our method can predict accurate Image-to-Point Cloud registrations, thus leads to a detailed and complete scene fusion results, where the fused scene before registration is a mess.

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

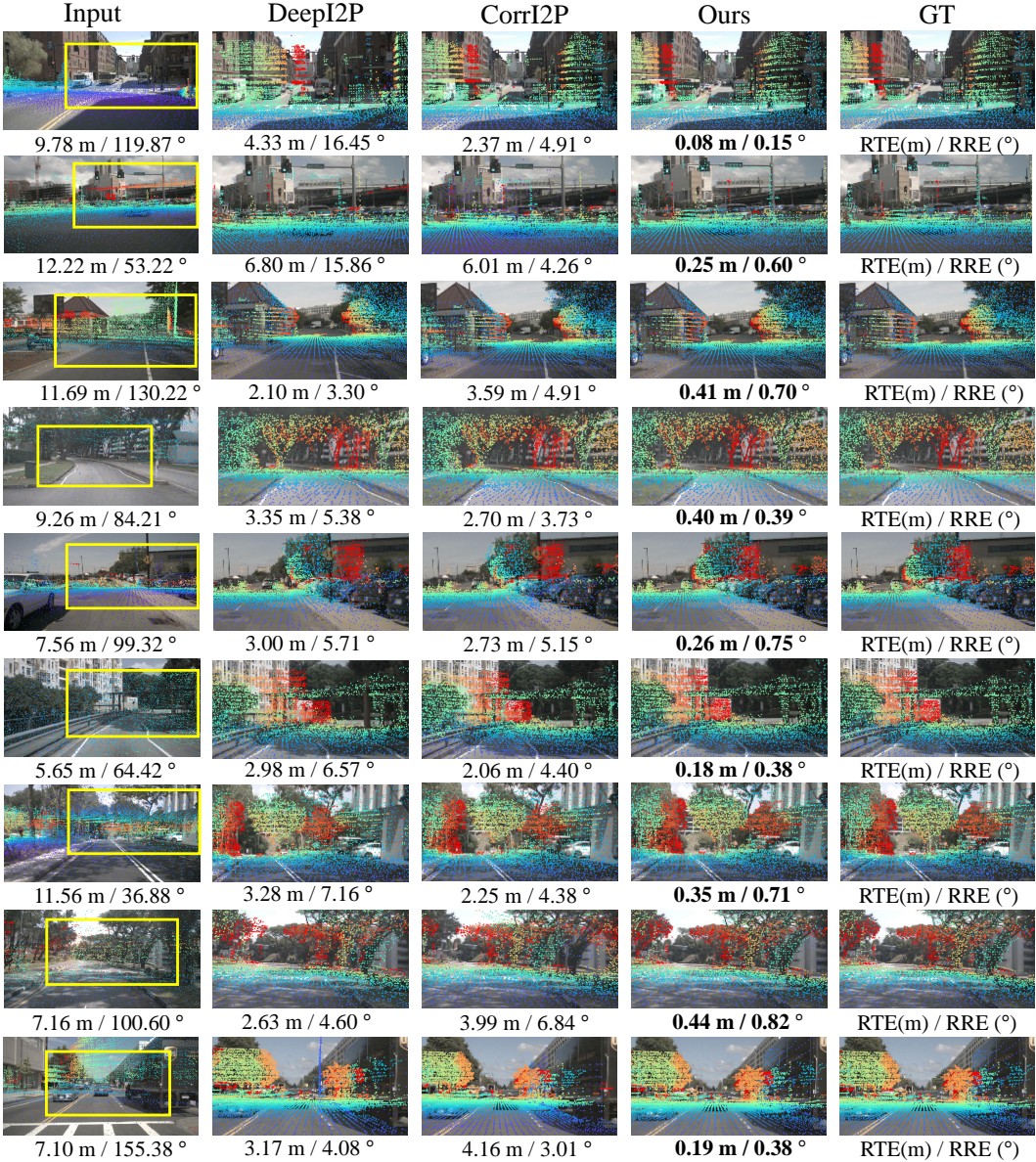

Figure 2: Registration comparison on nuScenes dataset. For visualization, the point cloud is projected into image space with the predicted extrinsic parameters of different approaches and the known camera internal parameters. The color indicates the distance between a point and the camera (close-to-far corresponds to blue-to-red).

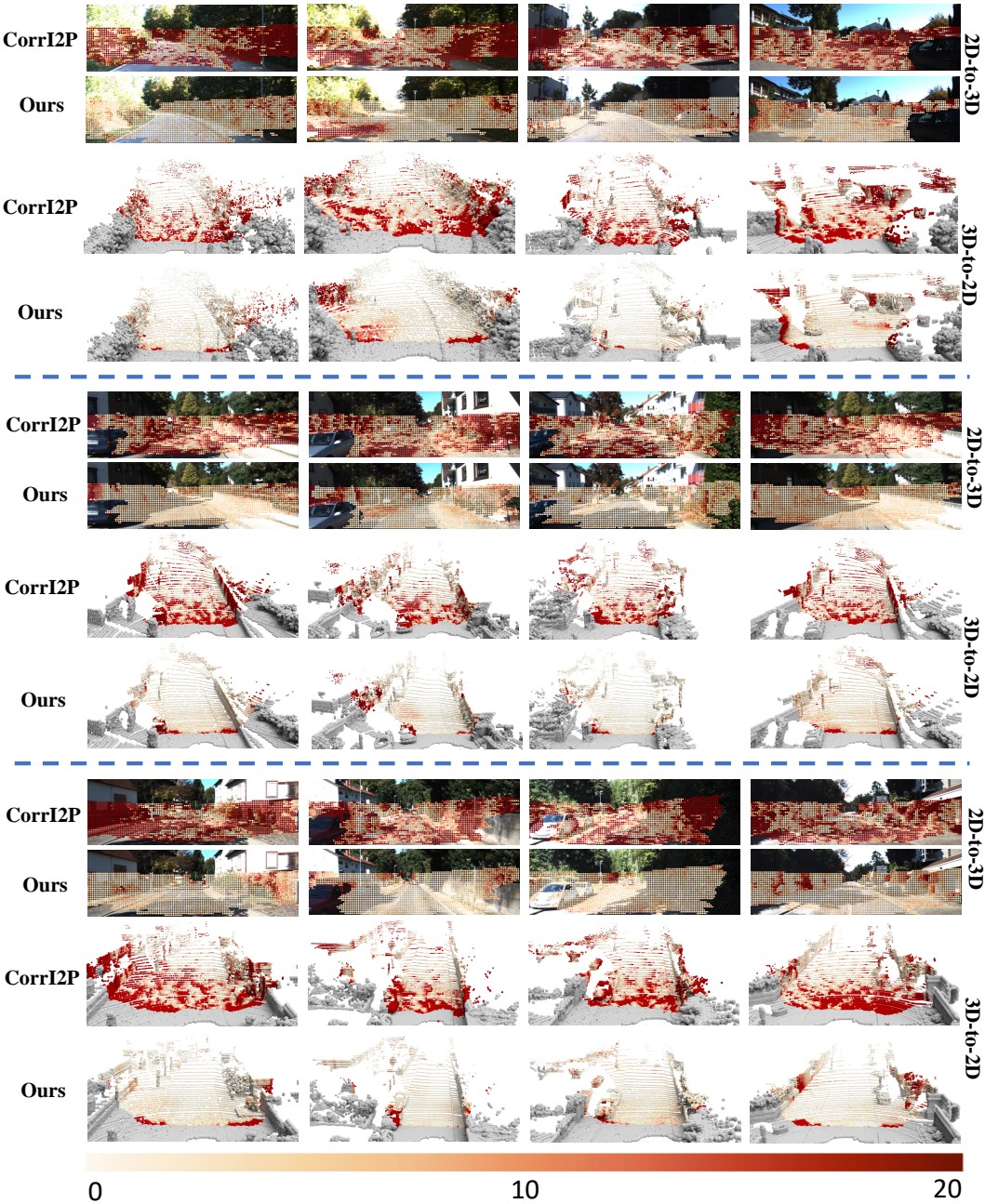

Figure 3: Feature matching comparison on KITTI dataset. For each pair, we visualize both the 2D-to-3D matching and 3D-to-2D matching errors of CorrI2P and our method. The colors indicate the error distances from 0 to 20 pixels, where the error larger than 20 pixels is set to 20 pixels for visualization. The gray color indicates the outlier regions in 3D.

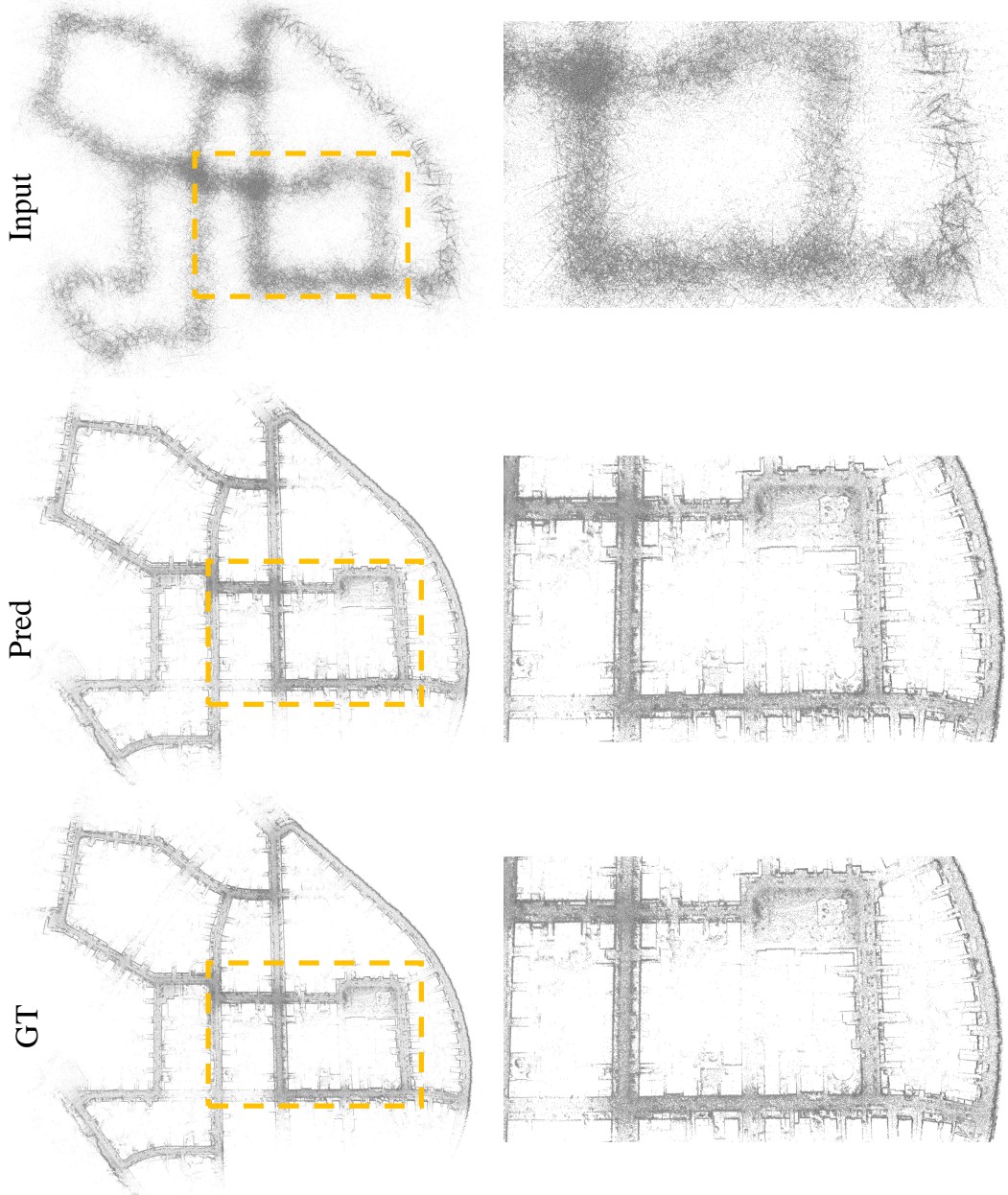

Figure 4: Scene fusion results on KITTI dataset. The 'Input' indicates the fused scene before registering LiDAR point cloud of each frame to the reference image in current frame. The 'Pred' indicates the fused scene with the Image-to-Point Cloud registration results to register LiDAR point cloud with the image in current frame using our method. The 'GT' indicates the fused scene with ground truth Image-to-Point Cloud transformations.