# OpenReview forum: "Differentiable Registration of Images and LiDAR Point Clouds with VoxelPoint-to-Pixel Matching"
_NeurIPS.cc/2023/Conference — NeurIPS 2023 spotlight_

### Official Review · Reviewer_Ehvn · 2023-07-03

**Soundness:** 3 good
**Presentation:** 3 good
**Contribution:** 3 good
**Rating:** 6
**Confidence:** 4

**Summary:**

This paper introduces a framework for registering 2D images and 3D point clouds. The framework consists of three branches, for processing 2D images, 3D point clouds and 3D voxel constructed from the original 3D point clouds. The fused features of 3D point clouds and 3D voxel are used to establish correspondences with the features of the 2D images, and a differentiable PnP solver is used to calculate the transformation for end-to-end training. Experiments are conducted on two public datasets, KITTI and nuScenes, and the proposed method outperforms several existing baseline methods.

**Strengths:**

1.The three-branch structure is novel for this task and seems effective in improving the registration performance.

2.The proposed method achieved much higher performance than existing methods in the experimental setting.

3.The paper is generally well written and easy to follow.

**Weaknesses:**

Major

1.My major concern is the practical meaning of the proposed work. I wonder if there is a real scenario where this kind of registration is essential. Why not establish the correspondences between 2D images and 3D point clouds by device calibration? If we calibrate a LiDAR and an image camera, which should not be difficult, we can register the data captured with it (including both images and point clouds) to either an image or a point cloud by mono-modal registration. This two kinds of mono-modal registrations are both well studied with much higher accuracy.

2.The ablation study has some issues. First, I notice that ablation is conducted on a different setting from the main experiment. Why not use the same setting as in Table 1? In the ablation study, only sequence 0-1 is used for training and only sequence 7 is used for testing, which are much less than the main experiment and makes the ablation results unstable. Second, from the ablation study we can see that removing any one component results in a light decrease of performance, but all ablation study performance are still much higher than baseline methods in Table 1. Existing ablation results can not show where the performance gain of the proposed method comes from. Further analysis is needed, such as removing two or three components at the same.

3.The data are split into training and testing sets, but the author did not mention if a validation set is used and how they chose the best model for inference.

4.RANSAC is used for more robust registration of the proposed method. We know that RANSAC plays a very important role in correspondence-based registration. Can RANSAC also be used in the comparing methods to improve their performance?

Minor

1.The references [31,32] and [19,22] in the first sentence of the second paragraph of Introduction are not proper, since these papers does not study the problem of establishing 2D-3D correspondences.

2.In the introduction of the KITTI datasets, “2D translation on the ground within ±10”, what’s the unit of the translation?

3.“registration accuracy (Acc.)” is not a good name for its current definition. Registration recall may be better.

4.The inference time is 0.19s, which can not be said as real-time.

**Questions:**

Please respond to questions raised in weakness.

**Limitations:**

limitation is not discussed.

---

> ### Author Rebuttal · Authors · 2023-08-09
>
> **Q1: Practical meaning of the proposed work.**
>
> There are two types of calibration between vehicle-mounted camera and LiDAR, which is pre-calibration and online-calibration, both of them play important roles in the autonomous driving systems. We focus on the online-calibration which is much harder but crucial for autonomous driving scenarios.
> The pre-calibration requires manual setup and adjustment, which brings significant labor costs. For example, the KITTI [1] dataset points out that the pre-calibration is not accurate where manual calibration is required to obtain the real transformation. As described in Sec. 2.2 of the KITTI [1] paper, the accurate transformation is achieved by first using the pre-calibration methods and then selecting a few manually correspondences between the LiDAR point clouds and pixels on the images to adjust the transformations, which is very labor-intensive. On the contrary, we aim to calibrate online automatically at any time during driving, which do not require any manual operation.
>
> Another main difficulty in calibration is the fact that the transformation between the vehicle-mounted camera and LiDAR is not constant during driving due to factors such as vehicle shake and rough road conditions. As a result, mis-calibration occurs in almost every frame, and the mis-calibration errors will continue to accumulate over time as the vehicle travels. The naïve pre-calibration methods that only calibrate once are unable to handle these mis-calibrations during driving and require manual adjustment every once in a while.
> Our method enables online calibration automatically with a fast inference of only 0.19s per frame, which can handle the mis-calibration errors at any time during driving and provide an intelligent and effective way for improving the robustness of autonomous driving systems.
>
> [1] Geiger, Andreas, et.al. Are we ready for autonomous driving? the kitti vision benchmark suite. CVPR2012
>
> **Q2: The data used for ablation studies.**
>
> We conduct ablation studies to explore each design in our frameworks and some important hyper-parameters like image resolutions, point densities, safe radius, feature dimensions, etc. Using a subset of the large-scale dataset for ablation studies is an efficient way and can justify the effectiveness of the designs.
> We leverage the first two sequence of KITTI dataset, which contains about 30% Image-to-Point Cloud pairs of the KITTI dataset, as a subset to conduct comprehensive ablations for efficiently verifying more hyper-parameters and designs.
> We agree that conducting ablation studies with the same setting as the main experiment to use the whole KITTI dataset can validate the design choices more thoroughly than only using a subset dataset. We will conduct the ablation studies under the whole KITTI dataset in the revision, due to the limited time in rebuttal period.
>
>
> **Q3: Comprehensive ablations on the designs.**
>
> We provide the comprehensive ablations on our framework designs in Table H of the rebuttal PDF. By removing all our designs on the Voxel branch, Adaptive-Weighted Optimization and Differential PnP, the performance degenerates from 0.65/2.10/91.14 to 1.98/4.73/61.50 in terms of RTE/RRE/Acc., which is a total failure. We also report the performance using one, two or three of our designs at the same in Table H of the rebuttal PDF, where the results comprehensively demonstrate the effectiveness of each of our design.
>
> **Q4: How to choose the best model?**
>
> We follow previous works (e.g. CorrI2P and DeepI2P) to split the data into training and testing sets for a fair comparison, where no validation set is used. We train our method for 50 epochs and use the final model for inference.
>
> **Q5: Can RANSAC improve other methods.**
>
> Actually, we already use the same setting as ours to apply EPnP with RANSAC to predict the rigid transformation at inference time for the SoTA baseline CorrI2P as described in L.283-L.285 of Sec. 4.2. The reason why CorrI2P fails to achieve accurate registrations is that CorrI2P produces lots of wrong matchings which results in wrong 2D-3D correspondences, and after lots of iterations，RANSAC can hardly eliminate the wrong correspondences. On the contrary, our method produces much better cross-modality matchings and leads to robust 2D-3D correspondences, thus can achieve accurate registrations where only a few iterations are required for RANSAC. The other baseline DeepI2P performs the frustum classification where no 2D-3D correspondences are conducted and is not suitable for EPnP with RANSAC to solve the final pose estimation. However, DeepI2P solves the inverse camera projection problem for pose estimation, with a 60-fold pose initialization for avoiding crashing, which plays a similar role as RANSAC in EPnP.
>
> **Q6: Minor problems.**
>
> 1. We will change the references [31, 32] and [19, 22] in the introduction to proper papers studying 2D-3D correspondences.
>
> 2. The unit of the translation in L.267 which introduces KITTI dataset is m (meter), we will correct the statement as “2D translation on the ground within ±10 m”.
>
> 3. We will change the name “Registration accuracy” to “Registration recall” in the full text following your suggestions.
>
> 4. We will correct the “real-time” statement into “fast registration with 5 fps”.

---

> > ### Comment · Reviewer_Ehvn · 2023-08-13
> > **comments on response**
> >
> > My concerns on this papere are well addressed in the response though some of them can only be included in further revision because of the time limit. Especially, I appriciate the detailed ablation study conducted in the response. I update my score to weak accept.

---

> > > ### Author Response · Authors · 2023-08-13
> > > **Thanks to reviewer Ehvn**
> > >
> > > Many thanks for all the helpful comments and positive assessment. We really appreciate reviewer Ehvn for upgrading the score.

---

### Official Review · Reviewer_ec8L · 2023-07-04

**Soundness:** 3 good
**Presentation:** 3 good
**Contribution:** 2 fair
**Rating:** 5
**Confidence:** 4

**Summary:**

This work aims to address the image-to-point cloud registration task. The authors propose a VoxelPoint-to-Pixel matching framework, which consists of three network branches dedicated to extracting features from voxel, point, and pixel representations, respectively, for 2D-3D matching. The network is trained with four different losses: an overlap prediction loss, a circle loss for 2D-3D feature matching, a KL divergence loss for probabilistic PnP, and a pose loss supervised by ground-truth. To assess the effectiveness of their approach, the authors perform experiments on two existing LiDAR datasets, showing improved 2D-3D matching performance compared to existing methods.

**Strengths:**

- The proposed 2D-3D matching framework combines several well-established techniques. To bridge the domain gap between points and pixels, the authors suggest employing a triplet network to extract element-wise 2D and 3D features. To enhance the robustness of the 2D-3D matching process, the authors incorporate an overlap prediction branch. The whole pipeline is supervised by a 2D-3D matching (circle) loss and pose estimation (PnP) losses.

- The authors assess the matching performance on the KITTI and nuScenes benchmarks, demonstrating that their method has better performance when compared with DeepI2P and CorrI2P.

- The paper is generally easy to comprehend. I appreciate the presence of suitable illustrations accompanying the explanations in Sec. 3.

**Weaknesses:**

- One main concern for this work is its limited technical innovations: the adaptive-weighted loss is basically the circle loss [39], while the KL divergence loss for probabilistic PnP is borrowed from Epro-PnP [8]. Additionally, there is a lack of ablation studies for those four losses - it is unclear how the KL divergence loss $L\_{KL}$ and the pose loss $L\_{pose}$ contribute to the matching performance and whether both of them are necessary.

- For the experiments on the KITTI and nuScenes datasets, the comparisons are limited to DeepI2P and CorrI2P. However, several other works related to 2D-3D matching are absent from the comparisons, such as the ones below. Additionally, it would be interesting to assess the generalization of the proposed method across datasets, for instance, from KITTI to nuScenes.
  - Hierarchical Scene Coordinate Classification and Regression for Visual Localization. CVPR 2020.
  - P2-Net: Joint Description and Detection of Local Features for Pixel and Point Matching. ICCV 2021.

- For the ablation study in Sec. 4.6, it is unclear why only sequences 0-1 are used. It would be more beneficial to include all available data in order to thoroughly validate the design choices.

- Typos:
  - L267, “mis-registration”?
  - L309 “dis-matched”?

**Questions:**

Please see the Weaknesses section.

**Limitations:**

The authors did not discuss the limitations of their method. It would benefit the reader to include a failure case analysis, which would provide a more comprehensive understanding of the proposed approach.

---

> ### Author Rebuttal · Authors · 2023-08-09
>
> **Q1: Technical innovations.**
>
> We did get inspiration from previous methods on some loss designs. However, to best of our knowledge, almost no work explored the cross-modality contrastive learning between image features and voxel-point features, where we design a triplet network to learn VoxelPoint-to-Pixel matching to reduce modality domain gap and lead to robust 2D-3D correspondences and registrations. Epro-PnP cannot be directly leveraged in the Image-to-Point Cloud task since the images and LiDAR point clouds are captured in quite different ways. There is a large range of outliers on both modalities where no correspondences can be founded. To handle the outlier regions, we design a detection strategy to predict the probability of lying in the intersection region for each 2D/3D elements, and remove the outlier regions on both modalities before inferring 2D-3D correspondences to solve Probabilistic PnP. Moreover, using the original contrastive equation, the network fails to produce a structured cross-modality latent space to represent both 2D and 3D features as shown in Figure 4. To address this issue, we introduce a loss with adaptive-weighted optimization inspired by Circle loss to explore a distinctive cross-modality latent space and also design a radius-based pair generation strategy to conduct negative and positive pairs, as discussed in Sec. 3.2. By directly introducing the original circle loss without our designed radius-based pair generation strategy, the network also fails to produce a structured cross-modality latent space to represent both 2D and 3D features, and the performance drops from 0.65/2.10/91.14 to 0.96/0.94/88.10 in terms of RTE/RRE/Acc.
>
> In all, we proposed the first end-to-end framework for learning Image-to-Point Cloud registration, which enables a fast inference. Compared to previous state-of-the-art methods, our framework reduces the registration error from 3.59 to 0.75 in terms of RTE. Additionally, it significantly reduces the inference time from 8.96s to 0.19s, demonstrating its promising capabilities in autonomous driving systems.
>
> **Q2: More ablations for designed losses.**
>
> We provide comprehensive ablation studies on each of the designed losses following your suggestions. We first demonstrate the effectiveness of our adaptive-weighted loss by comparing it with other metric learning losses (e.g. ContrastiveLoss, LiftedStructureLoss, GeneralizedLSLoss) in Table E of the rebuttal PDF, where we achieve the best performance among all the other losses. Then, we provide the ablations of replacing our Probabilistic PnP with another differential PnP BlindPnP [1] in Table F of the rebuttal PDF. We further provide the ablation studies to separately explore the impact of KL divergence loss and the pose loss in Table B of the rebuttal PDF, where both losses can improve the performance.
>
> [1] Campbell D, et.al. Solving the Blind Perspective-n-Point Problem End-To-End With Robust Differentiable Geometric Optimization. ECCV2020
>
> **Q3: Related works HSCNet and P2-Net.**
>
> We focus on difference tasks with the mentioned HSCNet and P2-Net. HSCNet focuses on the visual location task which aims to estimate the camera pose of a query image with respect to a known environment, as described in the first sentence of the introduction in HSCNet. We have distinguished our task with the visual location task in L.87-L.92 in the Related Work, where we focus on the image-to-point cloud registration task in dynamic scenes captured at different time steps instead of first pre-building the whole environment and learning to locate a query image. P2-Net learns pixel-point matching for establishing 2D-3D correspondences, and also focuses on the visual location task, which is a different task of ours as described above. We agree that the P2-Net which also learns 2D-3D feature matchings can be leveraged in the Image-to-Point Cloud registration task as ours with some modifications. However, P2-Net does not open source, where no implementations can be found for a fair comparison. Note that the GitHub link provided in the abstract of P2-Net is empty.
>
>
> **Q4: Generalize the proposed method across datasets.**
>
> We refer the reviewer to ”Global-Q1: Cross dataset validation.“ of the global response for justifying cross dataset validations.
>
>
> **Q5: The data used for ablation studies.**
>
> We conduct ablation studies to explore each design in our frameworks and some important hyper-parameters like image resolutions, point densities, safe radius, feature dimensions, etc. Using a subset of the large-scale dataset for ablation studies is an efficient way and can justify the effectiveness of the designs.
> We leverage the first two sequence of KITTI dataset, which contains about 30% Image-to-Point Cloud pairs of the KITTI dataset, as a subset to conduct comprehensive ablations for efficiently verifying more hyper-parameters and designs.
> We agree that conducting ablation studies with the same setting as the main experiment to use the whole KITTI dataset can validate the design choices more thoroughly than only using a subset dataset. We will conduct the ablation studies under the whole KITTI dataset in the revision, due to the limited time in rebuttal period.
>
>
> **Q6: Limitations and failure cases.**
>
> One of our limitations is that the feature matching errors at some noisy points of LiDAR point clouds may be very large, which have very negative influence on cross-modality registrations. As shown in Figure 3 of the supplementary, although the feature matchings at most of pixels/points are accurate, some feature matching results at noisy points (e.g. scans of bushes) of the scene are not stable. The reason is that the network is limited to handle noisy points without any special designs, leading to unstable 3D features at noisy points and further affect the registration accuracy at some complex scenes. We will add more discussion on the limitations and failure cases of our method in the revision.

---

> > ### Comment · Reviewer_ec8L · 2023-08-18
> > **Comments on Authors' Rebuttal**
> >
> > I appreciate the authors' effort in addressing most of my concerns, including technical contributions and loss ablations.
> >
> > One addtional comment, regarding the claim in the response, "almost no work explored the cross-modality contrastive learning between image features and voxel-point features", there is one seemingly related work I am aware of, though it is on transfer learning:
> > - Liu et al. Learning from 2d: Contrastive pixel-to-point knowledge transfer for 3d pretraining. 2021.
> >
> > Another question about the loss ablation study: were these new quantative results obtained on all available data of KITTI or still sequences 0-1?

---

> > > ### Author Response · Authors · 2023-08-19
> > > **Response to Additional Comments**
> > >
> > > Thanks for your response and we are happy to hear that our rebuttal helped. We would like to express our sincere gratitude for the constructive comments and valuable time. We respond to your additional questions as follows.
> > >
> > > **Q1: Claim in the response.**
> > >
> > > We do not claim to be the first to explore cross-modality contrastive learning, which was proposed in CorrI2P, P2-Net or the paper you mentioned [1] to leverage Point-to-Pixel Matching for learning cross-modality patterns. Actually, we claim to propose the first cross-modality contrastive learning framework between **image features and voxel-point features**, where we design a triplet network to learn **VoxelPoint-to-Pixel Matching**, instead of **Point-to-Pixel Matching**. It reduces the modality domain gap and leads to robust 2D-3D correspondences and registrations. As a comparison to the previous “Point-to-Pixel Matching” methods, we analyzed the effectiveness of voxels in “Comparison to Point-to-Pixel Matching” in Sec.3.1 and “The Analysis of VP2P Matching” in Sec.3.1 in the supplementary. We visualized the learned latent space of our VoxelPoint-to-Pixel Matching and quantitatively and qualitatively compared with Point-to-Pixel Matching to demonstrate that using regular voxels can reduce the domain gap between 2D and 3D data.
> > >
> > > Our motivation for introducing VoxelPoint-to-Pixel Matching for cross-modality contrastive learning is that irregular points are merely suitable to be processed by MLP to learn representations, while pixels are regular and processed by CNNs. The large differences between points and pixels and the one between calculations in MLPs and CNNs lead to different features domains and make it hard for previous “Point-to-Pixel Matching” works to learn a structured shared latent space for 2D and 3D data. We observe that voxels share much greater similarities with pixels than points since both voxels and pixels are regular and represented in grids, which are suitable to be processed by CNNs to operate spatially-local convolutions. Based on the analysis above, we propose **VoxelPoint-to-Pixel Matching** for reducing the 2D-3D modality domain gap, leading to robust and accurate Image-to-Point Cloud registrations.
> > >
> > > **Q2: Data for additional ablations in rebuttal.**
> > >
> > > For an intuitive comparison with the previous results, we conduct additional ablations under the same setting as Table 3 in Sec. 4.6 to use sequences 0-1 as the training data. Please note that we conducted extensive quantitative comparisons and ablation studies in the rebuttal, where we found it hard to conduct comprehensive ablations under the whole KITTI dataset due to the limited rebuttal period. We will conduct the ablation studies under the whole KITTI dataset in the revision.
> > >
> > >
> > > [1] Liu et al. Learning from 2d: Contrastive pixel-to-point knowledge transfer for 3d pretraining. 2021.

---

> > > > ### Comment · Reviewer_ec8L · 2023-08-19
> > > > **Comment on Authors' Reply**
> > > >
> > > > Thank you for the detailed and prompt reply. After considering the response and other review comments, I am willing to raise my rating to borderline accept to align with the fellow reviewers. In the revision, the authors should improve the ablation studies with full data as promised.

---

> > > > > ### Author Response · Authors · 2023-08-19
> > > > > **Thanks to Reviewer ec8L**
> > > > >
> > > > > Many thanks for all the helpful comments and positive assessment. We really appreciate reviewer ec8L for upgrading the score and will improve the ablation studies with full data in the revision.

---

### Official Review · Reviewer_yFFc · 2023-07-04

**Soundness:** 3 good
**Presentation:** 3 good
**Contribution:** 3 good
**Rating:** 6
**Confidence:** 4

**Summary:**

The paper proposes a 2D to 3D registration pipeline using a differentiable PnP method (Epro-PnP) and integrates 3D information using both voxel and point-based features. These designs target on previous problems such as the domain differences when fusing MLP-based point features and CNN-based pixel features, and the non-differentiable post-processing when computing the transformation that cannot train the entire model end-to-end. Experiments on KITTI and nuScenes, along with ablation studies show the effect of the proposed pipeline while achieving great efficiency.

**Strengths:**

- The proposed method improves the current state-of-the-art methods by a large margin in terms of the accuracy and efficiency on KITTI and nuScenes datasets.

- Although most techniques used in the paper are not new, combining these techniques yield effective results.

- The idea of combining both the voxel-based features and the point cloud-based features helps with the domain difference between MLP-based and CNN-based features.

- The writing of the paper is okay and is easy to follow.


**Weaknesses:**

Major:

- In the ablation study Table 3, the authors showed that the differentiable PnP solver improves the accuracy of the method. Could the authors further explain the reasons?

- In Line 287-288, the authors mentioned that they did not exclude extreme cases when reporting the performance. However, the authors could report both the performance with/without extreme cases for a clear and fair comparison to other methods.

- The paper uses non-differentiable PnP to solve pose during inference. What about other methods applying a fast PnP during inference? Will they achieve similar speedups?

- Ablation study (Table 3): how about using 1 or 2 of the components to show a more comprehensive study of the effectiveness of the combination of the module?

- Table 4 better adds efficiency comparison.

- The limitations of the method should be discussed in the main paper.



Minor:

- The attached code does not show many intuitions. A pseudo code is preferred in the text to aid the understanding of the method.

- There are some typos in the paper that need to attend to.


**Questions:**

Please address my questions/concerns above. Specifically, my main concerns are some unclear/inconsistent arguments and experimental designs.

**Limitations:**

The limitation of the method is not discussed. Some of the limitations that can be discussed are as follows:

1. The method uses a non-differentiable fast PnP solver during the inference, which is different from the optimization process. How will it affect the performance? And will any fast PnP solver be suitable here? What is the limitation during inference?

2. What is the limitation of the method when applied to different datasets/scenes?

3. How to choose model parameters in order to balance accuracy and efficiency? The authors have included some ablation studies in both the main paper and the supplementary materials. However, the tradeoff between accuracy and efficiency could be further discussed.

---

> ### Author Rebuttal · Authors · 2023-08-09
>
> **Q1: Why differentiable PnP can improve the accuracy?**
>
> The main insight that we introduce the end-to-end PnP is to impose supervisions directly on the predicted transformations. Previous works learn 2D-3D feature matching with non-differentiable PnP as a post-processing procedure to estimate the transformations The optimization target during training is only the pseudo supervision conducted from 2D-3D correspondences. The insufficient supervision leads to large errors since the network has no ability to handle incorrect matching pairs which have a highly negative effect on the results. By implementing an end-to-end framework, we are able to bring direct supervisions to the final transformation, which is the most accurate supervision and leads to a more stable and more accurate cross-modality registrations.
>
> **Q2: Comprehensive comparison with baseline methods.**
>
> We refer the reviewer to “Global-Q2: Comprehensive comparison with baseline methods. ” of the global response for the results and analyses.
>
> **Q3: Can fast PnP speedup other methods.**
>
> Actually, we already use the same setting as ours to apply EPnP with RANSAC to predict the rigid transformation at inference time for the SoTA baseline CorrI2P as described in L.283-L.285 of Sec. 4.2. The reason why CorrI2P fails to achieve fast pose inference is that CorrI2P produces lots of wrong matchings which results in wrong 2D-3D correspondences, and requires a large number of iterations for RANSAC to eliminate the wrong correspondences which is very time-consuming. On the contrary, our method produces much better cross-modality matchings and leads to robust 2D-3D correspondences as shown in Figure 6, thus only a few iterations are required for RANSAC to filter wrong correspondences. The other baseline DeepI2P performs the frustum classification where no 2D-3D correspondences are conducted and is not suitable for EPnP to solve the final pose estimation. Specifically, DeepI2P solves the time-consuming inverse camera projection problem for pose estimation, even with a 60-fold pose initialization for avoiding crashing, leading to long pose inference time. We also justify that our method requires much fewer parameters than the previous SoTA methods (e.g. CorrI2P and DeepI2P) as shown in Table 2, leading to much shorter time for network inference.
>
> **Q4: Comprehensive ablations on the designs.**
>
> We provide the comprehensive ablations on our framework designs in Table H of the rebuttal PDF. We also report the performance using one, two or three of our designs at the same in Table H of the rebuttal PDF, where the results comprehensively demonstrate the effectiveness of each of our design.
>
> **Q5: Add efficiency comparison to Table 4.**
>
> We add the efficiency comparison in Table 4 and provide the full results in Table G of in the rebuttal PDF.
>
> **Q6: Pseudo code for proposed method.**
>
> We provide the pseudo code for the training and testing progress of our propose method in Algorithm 1 and 2 in the rebuttal PDF.
>
> **Q7: What is the limitation during inference? How will leveraging EPnP for inference affect the performance? Will other fast PnP suitable here?**
>
> The main limitation during inference is the time spent, which determines the potential of our method to be applied in automatic driving scenarios where the low latency is required. We replace the Epro-PnP with EPnP during inference to reduce the pose inference time. We further test the performance of directly leveraging Epro-PnP to predict poses, which achieves similar performance (RTE = 0.74) with EPnP (RTE = 0.75) on the KITTI dataset. However, as reported in L.340-L.341, the inference time will increase from 0.19 s to 2.38 s if we use Epro-PnP. The reason is that Epro-PnP adopt a Gaussian-Newton algorithm-based iterative PnP solver with a time complexity of O($N^2$), while the EPnP is much faster with a time complexity of O($N$). Therefore, to enable fast registration, we leverage EPnP as the solver at the inference time. EPnP is the most widely-used and well-explored method for efficient pose estimation from correspondences with a time complexity of O(N), therefore we choose EPnP instead of other PnP solvers for fast and robust registration at inference time.
>
> **Q8: Limitations when applied to different datasets/scenes.**
>
> Our method shows great robustness and generality when getting applied in different datasets (e.g. KITTI and nuScenes), where we achieve more accurate registration results on both KITTI and nuScenes datasets as shown in Table 3 of the main paper. However, previous SoTA methods CorrI2P and DeepI2P show a performance decline in nuScenes. The result demonstrates that our method is more general than previous methods and achieves accurate performance in different dataset. We also made further discussions in the “KITTI vs. nuScenes” in Sec.3.2 of the supplementary.
> We further justify that in our experiments setting, each dataset is split among sequences which are collected in different scenes, and the test sequences are unseen scenes for the trained model.
> Therefore, our performance on KITTI or nuScenes dataset can already demonstrate the ability of our methods to generalize to unseen scenes.
>
> For more limitations of our method, please refer to “Global-Q3” and “Global-Q1” of the global respond.
>
> **Q9: How to choose model parameters for balancing accuracy and efficiency?**
>
> We mainly consider the accuracy when choosing model parameters and also take into account efficiency for the model lightweight and faster convergence. As shown in Table G of the rebuttal PDF, we observe that increasing image resolutions can greatly improve the performance (e.g. from 0.84/2.90 to 0.65/2.10 in terms of RTE/RRE), therefore we choose to use higher image resolution with acceptable computational cost. When increasing the point density, only a marginal improvement is achieved (e.g. from 0.71/2.01 to 0.60/2.09), therefore we choose a proper density with fine performance and efficiency.

---

> > ### Comment · Reviewer_yFFc · 2023-08-19
> > **Response to the authors**
> >
> > Thank the authors for providing additional information and experiments. I have read all the comments of the authors and the reviewers, and I appreciate that the authors could further add the ablation studies and the limitation discussions.
> >
> > Most of my concerns were addressed. I did not agree with the authors' response about the generalizability of the proposed method. However, the additional Table C may provide an example of the generalization of the method. The limitations I have listed were just some examples. I hope the authors could discuss this in the main text.
> >
> > In conclusion, I encourage the authors to improve the paper based on all the comments from the reviewers.

---

> > > ### Author Response · Authors · 2023-08-19
> > > **Thanks for the comments**
> > >
> > > Dear reviewer yFFc,
> > >
> > > We will follow your advice to update our manuscript. Thanks for your effort and time, and we really appreciate your expertise.
> > >
> > > Best,
> > >
> > > The authors

---

### Official Review · Reviewer_QyPU · 2023-07-05

**Soundness:** 3 good
**Presentation:** 3 good
**Contribution:** 2 fair
**Rating:** 5
**Confidence:** 5

**Summary:**

This paper proposes a method to register an image with its nearby Lidar scan, and it comes with three modules:

1) A sparse 3D conv-net and point-net to extract 3D features; A 2D conv-net to extract 2D features;

2)  An intersection detection module to discard non-matchable 2D and 3D points;

3) Modified Circle, Probabilistic PnP, and pose losses are used to train the proposed triple network.

**Strengths:**

1) This paper is well-written and easy to follow;

2) This paper makes good practice to combine multiple existing modules to address the image2Lidar registration problem;

3) According to Table 1, the proposed method outperforms previous works addressing the image2Lidar registration problem.


**Weaknesses:**

Though this paper successfully combines multiple existing modules, some design choices are yet to be investigated. Specifically,

1) Generalization ability.

According to the paper, the proposed method focuses on autonomous driving datasets and conducts experiments on 3-DoF registration problem (Line 267). I have a concern that the network overfits on this specific configuration. To address my concern, please conduct the following experiments:

a) Cross-validation. Using a network trained on the KITTI dataset to test on the nuScenes dataset;

b) Conducting 6-DoF registration, rather than 3-DoF registration. Please at least add rotations around the x-axis and y-axis;


2) Pipelines.

a) The effectiveness of the modified Circle Loss. Please add the comparison with respect to the original Circle Loss in Table 3;

b) Since metric-learning plays an important role in this paper, I would expect authors to conduct comparisons with off-the-shelf metric-learning losses other than the Circle Loss. Please refer to https://github.com/KevinMusgrave/pytorch-metric-learning;

c) The effectiveness of the Probabilistic PnP and pose loss. I would expect a two-stage training strategy here by first training the network by only using the adaptive-weighted loss, and then adding the Probabilistic PnP and pose loss. This reminds me of the work [Solving the Blind Perspective-n-Point Problem End-To-End With Robust Differentiable Geometric Optimization]. Please separate the impact of KL divergence loss and pose loss, and check their effectiveness independently in Table 3. It would be nice to further compare the proposed Probabilistic PnP with respect to the differentiable PnP module in [Solving the Blind Perspective-n-Point Problem End-To-End With Robust Differentiable Geometric Optimization].


3) Minor.

I think processing 2D and 3D data with convolutions may not justify the claim of "Sharing similar characteristics in feature space" (Line 162), as 3D convolution is invariant to shift, viewpoint, distance, etc. In contrast, 2D convolution is vulnerable to viewpoint and scale differences.


**Questions:**

Please refer to the weaknesses.

---

> ### Author Rebuttal · Authors · 2023-08-09
>
> **Q1: Cross dataset validation.**
>
> We refer the reviewer to ”Global-Q1: Cross dataset validation.“ of the global response for justifying cross dataset validations.
>
> **Q2: Conducting 6-DoF registration, rather than 3-DoF registration.**
>
> We follow previous methods (e.g. CorrI2P and DeepI2P) to conduct 3-DoF registration for a fair comparison. However, we justify that our proposed method is designed to match cross-modality features for image-to-point cloud registration and is not limited to the certain 3-DoF problem. We conduct 6-DoF registration under KITTI dataset to demonstrate the ability of our method in handling more difficult situations. Specifically, we conduct 6-DoF mis-registration transformation with 3D translations on x-axis, y-axis, and z-axis within $\pm$ 5 m, and rotations around the x-axis, y-axis and z-axis within $\pm$ 120$^o$. We report our performance and make a comparison with previous SoTA method CorrI2P in Table D in the rebuttal PDF. The results demonstrate that our method can also work in 6-DoF image-to-point cloud registration task with large mis-registrations where we achieve accurate registration performance of 0.96/3.73/76.7 in terms of RTE/RRE/Acc., while previous SoTA method CorrI2P fails at the more difficult task and produces much worse results (4.29/12.28/42.53) than ours.
>
> **Q3: Comparison with original Circle loss.**
>
> We provide the comparison with original Circle loss in “Original CircleLoss” in Table E in the rebuttal PDF, where the performance drops from 0.65/2.10/91.14 to 0.96/2.94/88.10 in terms of RTE/RTE/Acc without our designs on the adaptive weighted optimization.
>
> **Q4: Comparisons with off-the-shelf metric-learning losses.**
>
> We make a comparison with some off-the-shelf metric-learning losses in Table E in the rebuttal PDF. The implementation of these losses are from the GitHub repo as you suggested. The results demonstrate the effectiveness of our proposed adaptive-weighted optimization where we achieve the best performance among all the other losses (e.g. ContrastiveLoss, LiftedStructureLoss, GeneralizedLSLoss). The reason is that other metric-learning losses treat each pair of samples equally and are unable to distinguish hard and easy pairs, which leads to ambiguous convergence, especially in the difficult cross-modality matching. While we design a flexible optimization strategy with adaptive weighting to force the network to focus more on the harder samples, leading to a distinctive cross-modality latent space where we can establish 2D-3D correspondences more accurately. We further justify that our adaptive-weighted strategy is a general term that can be leveraged in improving other metric-learning losses, where we integrate the strategy to GeneralizedLSLoss and report the performance as “GeneralizedLSLoss-AW”.
>
>
> **Q5: The effectiveness of the Probabilistic PnP and pose loss.**
>
> We provide the ablation studies on Probabilistic PnP and pose loss in Table F of the rebuttal PDF, following your suggestions. We first report the performance of a two-stage training strategy as “TwoStage (w/ Ours PnP)” by first training the network only using the adaptive-weighted loss for half epochs and then adding the Probabilistic PnP losses for the rest epochs. The two-stage strategy outperforms the result of using only adaptive-weighted optimization as “w/o Diff. PnP” in Table F (0.69 vs. 0.75) and is slightly worse than the result of our one-stage training with Probabilistic PnP shown as “w/ Ours PnP” in Table F (0.69 vs. 0.65). We further provide the ablation studies to separately explore the impact of KL divergence loss and pose loss as shown in Table B of the rebuttal PDF, where both losses can improve the performance.
>
> **Q6: Compare Probabilistic PnP with BlindPnP.**
>
> We provide the ablations of replacing our Probabilistic PnP with BlindPnP [1] in Table F of the rebuttal PDF, where BlindPnP performs worse than our Probabilistic PnP (0.73 vs. 0.65). By introducing supervisions on the predicted pose distribution, the Probabilistic PnP brings more robust guidance for optimizations than BlindPnP which only leverages the L2 loss to guide the pose learning.
>
> [1] Campbell D,  et.al. Solving the Blind Perspective-n-Point Problem End-To-End With Robust Differentiable Geometric Optimization. ECCV2020
>
> **Q7: Can processing 2D and 3D data with convolutions reduce the domain gap?**
>
> Our motivation is that irregular points are merely suitable to be processed by MLP to learn representations, while pixels are regular and processed by CNNs. The large differences between points and pixels and the one between calculations in MLPs and CNNs lead to different features domains and make it hard for previous works to learn a structured shared latent space for 2D and 3D data.
> We observe that voxels share much greater similarities with pixels than points since both voxels and pixels are regular and represented in grids, which are suitable to be processed by CNNs. We agree that 2D and 3D CNNs also have some differences, but they share similar operation to perform convolutions on regular data (pixels or voxels) represented in grids to explore spatially-local patterns, which is quite different to the feature patterns obtained by performing MLPs in irregular point sets. We analyzed the effectiveness of introducing voxel information in “Comparison to Point-to-Pixel Matching” of Sec.3.1 and “The Analysis of VP2P Matching” of Sec.3.1 in the supplementary. We visualized the learned latent space of our VoxelPoint-to-Pixel Matching and quantitatively and qualitatively compared with Point-to-Pixel Matching to demonstrate that using regular voxels can reduce the domain gap between 2D and 3D data.

---

> > ### Comment · Reviewer_QyPU · 2023-08-17
> >
> > Thanks for the rebuttal and new experiments.
> >
> > They addressed my comments well, and I would update my score to borderline accept.
> >
> > Please reflect these new experiments in the revised paper.

---

> > > ### Author Response · Authors · 2023-08-17
> > > **Thanks to Reviewer QyPU**
> > >
> > > Dear Reviewer QyPU,
> > >
> > > Many thanks for all the helpful comments and positive assessment, we will add the new experiments in the revised paper. We really appreciate you for upgrading the score.
> > >
> > > Best,
> > >
> > > Authors

---

### Official Review · Reviewer_CqXY · 2023-07-06

**Soundness:** 3 good
**Presentation:** 3 good
**Contribution:** 3 good
**Rating:** 6
**Confidence:** 4

**Summary:**

The paper proposes to learn a structured cross-modality latent space to represent pixel features and 3D features via a differentiable probabilistic PnP solver, which designs a triplet network to learn VoxelPoint-to-Pixel matching. The proposed method is trained in end-to-end manner by imposing supervisions directly on the predicted pose distribution with a probabilistic PnP solver. The experiments seem good.

**Strengths:**

(1). A framework to learn Image-to-Point Cloud registration by learning a
structured cross-modality latent space with adaptive-weighted optimization, together with
an end-to-end training schema driven by a differentiable PnP solver.

(2) The paper represents the 3D elements as the combination of voxels and points to
overcome the pattern gap between points and pixels, where a triplet network is designed to learn VoxelPoint-to-Pixel matching.

**Weaknesses:**


1. The motivation that use voxel information is unclear, as described in the paper, one of the bottleneck of previous methods is points and pixels are with different characteristics with patterns learned in different manners (MLP and CNN), however, domain gap is also exited between voxeled information and pixels, why does the paper use voxel information should be discussed.

2. The ablation studies are not enough,
(1). in 3.2 Adaptive-Weighted Optimization, a hyper-parameter radius r is used, which is should be discussed, because it affects the positive and negative pairs, experiments should be provided to verify the robustness.

(2). In probabilistic PnP, the effectiveness of L_{pose} and L_{kl} should be verified separately.

(3). I agree that the settings in CorrI2P is unsuitable that exclude RTE larger than 5m and RRE larger than 10◦, however, it is better to provide these results same with CorrI2P, since it would be better for readers to understand where does main improvement come from.

(4). Limitatations and failure cases should be discussed.


**Questions:**

Please see weaknesses.

**Limitations:**

Limitations and failure cases should be discussed.

---

> ### Author Rebuttal · Authors · 2023-08-09
>
> **Q1: Motivation of introducing voxel branch.**
>
> Our motivation is that irregular points are merely suitable to be processed by MLP to learn representations, while pixels are regular and processed by CNNs. The large differences between points and pixels and the one between calculations in MLPs and CNNs lead to different features domains and make it hard for previous works to learn a structured shared latent space for 2D and 3D data. We observe that voxels share much greater similarities with pixels than points since both voxels and pixels are regular and represented in grids, which are suitable to be processed by CNNs to operate spatially-local convolutions. We analyzed the effectiveness of introducing voxel information in “Comparison to Point-to-Pixel Matching” of Sec.3.1 and “The Analysis of VP2P Matching” of Sec.3.1 in the supplementary. We visualized the learned latent space of our VoxelPoint-to-Pixel Matching and quantitatively and qualitatively compared with Point-to-Pixel Matching to demonstrate that using regular voxels can reduce the domain gap between 2D and 3D data.
>
>
> **Q2: Ablations on hyper-parameter radius $r$.**
>
> We have provided the ablation studies on the safe radius $r$ in Sec.3.3 in the supplementary. We set the safe radius $r$ to 0.5, 1, 2 and 4 pixels, and report the performances in Table 2 of the supplementary.
>
>
> **Q3: The effectiveness of $L_{pose}$ and $L_{KL}$.**
>
> We conducted the ablations to verify $ L_{KL}$ and $ L_{pose}$ separately in Table B in the rebuttal PDF. We observe that when removing the probabilistic PnP loss $ L_{KL}$ in Eq. (7) or the pose loss $L_{pose}$ in Eq. (8) separately, the RTE (lower is better) rise from 0.65 to 0.72 / 0.68. And when further removing both of them, the RTE further degenerates to 0.75. These results demonstrate that the probabilistic PnP loss in Eq. (7) brings major improvement to the registration accuracy and the pose loss in Eq. (8) provides additional enhancements.
>
>
> **Q4: Comprehensive comparison with baseline methods.**
>
> We provide the comprehensive comparison with baseline methods (e.g. CorrI2P and DeepI2P) under the same setting of CorrI2P to exclude RTE larger than 5m and RRE larger than 10$^o$ in Table A of the rebuttal PDF.
> To further provide a more real and convincing comparison with SoTA method CorrI2P in the performance without extreme cases, we remove the same number of bad samples as CorrI2P and report the performance of our method as “Ours *”. By evaluating on the same number of test samples without extreme cases, we believe it is a relative fair comparison with CorrI2P. As shown in Table A of the rebuttal PDF, we achieve the best performance under all metrics. Especially, our method is about 3 times better than the SoTA baseline CorrI2P under the difficult benchmark nuScenes, even after removing the extreme samples. The result demonstrates that our method not only produces more stable registrations with much fewer failure cases but also produces much precise registrations for the success samples.
>
>
> **Q5: Limitations and failure cases.**
>
> One of our limitations is that the feature matching errors at some noisy points of LiDAR point clouds may be very large, which have very negative influence on cross-modality registrations. As shown in Figure 3 of the supplementary, although the feature matchings at most of pixels/points are accurate, some feature matching results at noisy points (e.g. scans of bushes) of the scene are not stable. The reason is that the network is limited to handle noisy points without any special designs, leading to unstable 3D features at noisy points and further affect the registration accuracy at some complex scenes. We will add more discussion on the limitations and failure cases of our method in the revision.

---

> > ### Comment · Reviewer_CqXY · 2023-08-17
> >
> > Thanks for the rebuttal, most of our concerns are addressed. Though the motivations should be further discussed in the following versions of the paper, the experiments can prove the proposed point to some extent, so I update my score to weak accept.

---

> > > ### Author Response · Authors · 2023-08-17
> > > **Rating clarification inquiry**
> > >
> > > Dear Reviewer CqXY,
> > >
> > > We would like to express our sincere gratitude for the constructive comments you provided on our work. Your insights are invaluable, and we are committed to incorporating your suggestions and will further discuss the motivations in the following versions of the paper.
> > >
> > > We do, however, have a slight query that we hope you can kindly clarify. We noted that your previous score was already "borderline accept". In light of your recent positive feedback, are you suggesting to "update the score to weak accept" instead of "update the score to borderline accept"? Your clarification would be greatly appreciated.
> > >
> > > Thank you once again for your thoughtful feedback and time invested in evaluating our work.
> > >
> > > Best regards,
> > >
> > > Authors

---

> > > > ### Comment · Reviewer_CqXY · 2023-08-17
> > > >
> > > > Thank you for the information, I have fixed it and update to weak accept.

---

> > > > ### Comment · Reviewer_QyPU · 2023-08-17
> > > >
> > > > Dear author,
> > > >
> > > > I have already updated the score from borderline reject to borderline accept. I have edited my initial borderline reject rating directly.
> > > >
> > > > Best,
> > > > Reviewer.

---

### Author Rebuttal · Authors · 2023-08-09

We upload a rebuttal PDF with some experimental results requested by the reviewers. For the following rebuttals, we use “rebuttal PDF” to point to the provided PDF like “in Table A of the rebuttal PDF”.

We respond to some common questions in reviews as follows.

**Global-Q1: Cross dataset validation.**

Almost all the widely-used methods (e.g. Cylinder3D[1], CenterPoint[2])in 3D perception (e.g. segmentation / detection) train a single model for a dataset, i.e. one model for KITTI and one model for nuScenes, and do not perform cross dataset validation. The reason is that cross dataset validation is extremely difficult for 3D perception tasks of autopilot, since the devices (e.g. LiDARs and cameras) and the ways to collect data are quite different from dataset to dataset (e.g. KITTI and nuScenes). This leads to large differences of predicted feature distributions when apply a model trained on one dataset to directly leverage it to evaluate on another dataset. However, we justify that the experiments under a single dataset can already demonstrate the ability of our methods to generalize to unseen sequences or scenes. Specifically, each dataset (e.g. KITTI, nuScenes) is split among sequences and different sequences are collected in different scenes, therefore the test sequences are unseen scenes for the trained model.

Moreover, we justify that our trained model in a single dataset can learn some underlying patterns that can be shared across different datasets. Specifically, we leverage our pretrained model of KITTI dataset to conduct few-shot registration experiments under nuScenes dataset. The model is finetuned using only 10 % samples in nuScenes training set, and is evaluated on the full test set. The result is shown in “$Ours_{Scratch}$” and “$Ours_{KITTIPretrain}$” in Table C of the rebuttal PDF, where fine-tuning the pretrained model of KITTI dataset on the small subset of nuScenes dataset significantly outperforms the result of training a randomly initialized model from scratch ,i.e. 1.79 vs. 2.72 in terms of RTE.

[1] Zhu X, et al. Cylindrical and asymmetrical 3d convolution networks for LiDAR segmentation. CVPR2021

[2] Yin T, et.al. Center-based 3d object detection and tracking. CVPR2021

**Global-Q2: Comprehensive comparison with baseline methods.**

We provide the comprehensive comparison with baseline methods (e.g. CorrI2P and DeepI2P) under the same setting of CorrI2P to exclude RTE larger than 5m and RRE larger than 10$^o$ in Table A of the rebuttal PDF.
To further provide a more real and convincing comparison with SoTA method CorrI2P in the performance without extreme cases, we remove the same number of bad samples as CorrI2P and report the performance of our method as “Ours *”. By evaluating on the same number of test samples without extreme cases, we believe it is a relative fair comparison with CorrI2P. As shown in Table A of the rebuttal PDF, we achieve the best performance under all metrics. Especially, our method is about 3 times better than the SoTA baseline CorrI2P under the difficult benchmark nuScenes, even after removing the extreme samples. The result demonstrates that our method not only produces more stable registrations with much fewer failure cases but also produces much precise registrations for the success samples.

**Global-Q3: Limitations and failure cases.**

One of our limitations is that the feature matching errors at some noisy points of LiDAR point clouds may be very large, which have very negative influence on cross-modality registrations. As shown in Figure 3 of the supplementary, although the feature matchings at most of pixels/points are accurate, some feature matching results at noisy points (e.g. scans of bushes) of the scene are not stable. The reason is that the network is limited to handle noisy points without any special designs, leading to unstable 3D features at noisy points and further affect the registration accuracy at some complex scenes. We will add more discussion on the limitations and failure cases of our method in the revision.

---

### Author Response · Authors · 2023-08-10
**Response to all: Thank you very much for the thorough reviews.**

We deeply appreciate the reviewers for their thoughtful feedback and time invested in evaluating our work. We are delighted that reviewers appreciated the representation and the significance of the paper. We are heartened by Reviewer Ehvn's commendation on the novelty of our method and by both Reviewer CqXY and Reviewer ec8L's acknowledgement on the capability of our work to "bridge the domain gap between points and pixels". We are also pleased that the Reviewer yFFc found our method "improves the current state-of-the-art methods by a large margin". We are excited by Reviewer QyPU‘s characterization of our work as "makes good practice". Moreover, we are encouraged with the recognition of reviewers that our paper is "well-written", "easy to follow" and "with suitable illustrations accompanying the explanations".

We make a global response with a rebuttal PDF containing some experimental results and pseudo codes requested by the reviewers. We further respond to each reviewer separately. Thank you again for your invaluable feedback and we are looking forward to continuing the discussion.

---

### Decision · Program_Chairs · 2023-09-21

**Decision:**

Accept (spotlight)

**Comment:**

This paper received unanimously positive reviews, including three weak accepts and two borderline accepts. All the reviewers recognized the effective design of the proposed approach and the state-of-the-art performance on standard benchmarks.

The AC thus recommends to accept it as a Spotlight.